# Bio-Vision-Inspired Spiking Neural Networks for Object Detection with Event Cameras

**Dongyang Ma** [1]  **Zhengyu Ma** [2]  **Yifan Huang** [1]  **Chenlin Zhou** [2 3]  **Wei Zhang** [2]  **Yonghong Tian** [1 2 3]

## Abstract

Retina-like event cameras and brain-inspired Spiking Neural Networks (SNNs) demonstrate exceptional energy efficiency through bio-inspired sensing and computation. While SNNs are naturally well-suited to the asynchronous nature of event data, their practical applications face the following challenges: sensitivity to noise, dense representations that disrupt spike pathways, and insufficient multi-scale feature perception. To address the aforementioned challenges, we propose a bio-vision-inspired object detection method motivated by biological (bio) vision systems. First, at the micro level, this paper proposes a noise-filtering STATNF-Neuron architecture to address the current sensitivity of basic neurons to noise. Based on STATNF-Neurons, the paper introduces two bio-vision-inspired macro-structures: Events-to-Spikes Representation (E2S), which preserves spiking characteristics while mimicking the memory and noise-filtering abilities of retinal neurons; Bidirectional Multi-Scale Spiking Network (BiSNet), which simulates cortical information flow pathways to integrate multi-scale features in both directions, enhancing the network's ability to perceive information at multiple scales. Extensive experiments show that the proposed bio-vision-inspired method achieving state-of-the-art performance. Notably, it reaches 96.1% accuracy on NCAR, 63.5% mAP$_{50}$ on N-Caltech101, and 69.1% mAP$_{50}$ on Gen1.

[1]School of Computer Science, Peking University, Beijing, China [2]Peng Cheng Laboratory, Shenzhen, China [3]School of Electronic and Computer Engineering, Peking University, Shenzhen, China. Correspondence to: Zhengyu Ma <mazhy@pcl.ac.cn>, Yonghong Tian <yhtian@pku.edu.cn>.

*Proceedings of the 43$^{rd}$ International Conference on Machine Learning*, Seoul, South Korea. PMLR 306, 2026. Copyright 2026 by the author(s).

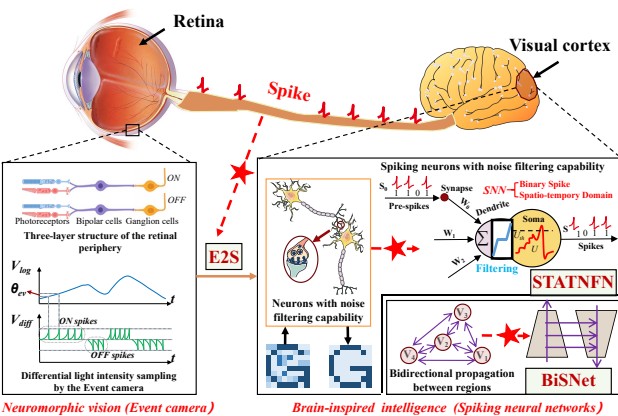

*Figure 1.* Biological visual systems, inspiring neuromorphic vision sensors and brain-inspired computing paradigms. Inspired by neurons' noise filtering, we propose the spiking neuron with noise filtering (STATNFN). Inspired by retinal denoising, motion perception, memory, multi-scale perception, and spike retention, we propose Events-to-Spikes Representation (E2S). Inspired by bidirectional information transmission across visual cortex, we propose Bidirectional Multi-Scale Spiking Network (BiSNet).

## 1. Introduction

Object detection plays a crucial role in computer vision, with key applications including autonomous systems (Li et al., 2019), medical imaging (Ma et al., 2020), remote sensing (Gao et al., 2023), robotics (Karaoguz & Jensfelt, 2019) and multi-target tracking (Zhang et al., 2022). The main drawbacks of traditional frame-based and deep Artificial Neural Networks (ANNs) object detection lie in their vulnerability to high-speed motion and extreme lighting, high data redundancy, and high power consumption (Li et al., 2022; Liu et al., 2020; Sayed & Brostow, 2021). Achieving object detection on embedded devices while ensuring computational efficiency and critically has remained a fundamental research challenge (Li et al., 2025; 2022).

The biological visual system, comprising the retina and visual cortex, operates with remarkable efficiency (Cai et al., 2023). While exhibiting extraordinary cognitive capabilities, the human brain consumes only about 20 W of power, far surpassing electronic computers in energy efficiency (Liang et al.). This biological advantage has inspired the development of neuromorphic vision sensors and brain-inspired computing (Figure 1). Event cameras (Lichtsteiner et al.,

2008), as quintessential neuromorphic sensors, replicate ON/OFF-type neural responses through device-level emulation of the retinal signal pathway (photoreceptors → bipolar cells → ganglion cells). These sensors achieve superior dynamic range, microsecond temporal resolution, and ultra-low power consumption (Peng et al., 2023a; Wang et al., 2023a), making them ideal for challenging scenarios (Peng et al., 2023a;b; Wang et al., 2023a; 2024; Zubic et al., 2024; Yuan et al., 2024; Hamaguchi et al., 2023; Zubić et al., 2023; Tomy et al., 2022). SNNs (Maass, 1997), representing third-generation neural networks, implement event-driven computation via spatiotemporal spike patterns, enabling high-fidelity simulation of neuronal dynamics with biological plausibility (**Appendix A.1**).

As shown in (Li et al., 2022), combining event cameras with SNNs serves as an effective paradigm for constructing energy-efficient object detectors. However, studies such as (Yan et al., 2025) have revealed significant limitations in current SNNs for processing complex temporal signals, notably their insufficient robustness to noise and weak multi-scale modeling capabilities. In contrast, biological visual systems can effectively handle such challenges. However, most existing approaches lack exploration of bio-vision-inspired mechanisms and face the following challenges:

The **noise-handling capability** and spike firing rate of neurons in SNNs are critically important for both accuracy and energy consumption. Noise not only degrades feature quality but also triggers unnecessary spikes, subsequently elevating energy consumption. In contrast, neuronal activity in the cerebral cortex exhibits high sparsity (Yan et al., 2025), with typically <5% of neurons active. Visual neurons demonstrate intrinsic noise-filtering capabilities (Xi et al., 2024). However, biological properties such as the low firing rates and noise suppression observed in visual neurons have not yet been adequately explored in current research on event data and SNN-based object detection.

Most existing event data and SNN-based detection methods rely on **accumulation-based representations** like HIST (Su et al., 2023), VoxelCube (Cordone et al., 2022), or MESTOR (Mao et al., 2025), which bin asynchronous events into discrete frames (**Appendix A.2**). From a neuromorphic perspective, these representations disrupt the high-fidelity spike-based transmission in the pathway from the retina to visual cortex (Figure 1). Algorithmically, the accumulation-induced non-binary inputs break end-to-end spiking characteristics, diminishing neuromorphic advantages. Moreover, these accumulation operations tend to enhance temporal noise, introduce redundant information, and overlook temporal dependencies between frames, thereby breaking spatiotemporal continuity.

The challenges of **multi-scale perception** continue to impact SNN-based event object detection. Prior efforts have attempted to mitigate these problems, but with limited success. (Li et al., 2022) proposes using different SNN layers for prediction. (Su et al., 2023) implements complex architectures such as membrane potential residual pathways, yet these designs retain noise and amplify feature redundancy, increasing energy consumption. (Fan et al., 2024) introduced feature fusion networks to enhance detection accuracy through multi-scale integration, yet the improvements remain constrained.

To address the aforementioned challenges, we propose STATNF-BiSNet from a bio-vision-inspired perspective, a co-designed Bidirectional Multi-Scale Spiking Network with Spatio-Temporal Adaptive Threshold Noise Filtering Neurons (STATNFN) for event-based object detection. First, inspired by the spatiotemporal feature processing and noise-filtering capabilities of visual neurons, we develop a noise-filtering neuron architecture. Its core innovation is a Spatio-Temporal Adaptive Threshold Noise Filtering mechanism integrated into the base neuron model, enabling simultaneous noise suppression and spike reduction. Building on STATNFN and guided by bio-vision systems, we introduce two macro-structures: (1) Events-to-Spikes Representation (E2S) preserves end-to-end spiking decoding and emulates retinal neurons' memorization, motion recognition, and noise-filtering functions. (2) Bidirectional Multi-Scale Spiking Network (BiSNet) models bidirectional information transmission across visual cortices (V1-V4) (Xin et al., 2025), enabling each network layer to integrate fine-grained details and high-level semantics for efficient multi-scale feature fusion. Experimental results demonstrate that after incorporating the bio-vision-inspired mechanisms proposed in this paper, a significant improvement in accuracy (mAP increased by 7.7%) is achieved with only a modest increase in parameters (+1.1M). Our main contributions are:

- **STATNF-Neuron:** We design a neuron architecture inspired by the noise-filtering capabilities and sparse spiking properties of bio-visual neurons. The proposed STATNF mechanism dynamically regulate membrane potential, enabling fundamental neurons to filter noise while significantly reducing SNN spike rates without compromising model accuracy.

- **E2S Representation:** We design an events representation method that simulates retinal functions. This method preserves end-to-end spiking properties in Event-SNN architectures while emulating the integrated capabilities of retinal neurons for memorization, motion recognition, and noise filtering.

- **BiSNet:** Our architecture models bidirectional information transmission across visual cortices, enabling each network layer to jointly integrate fine-grained details and high-level semantics for efficient multi-scale feature fusion without additional neck networks.

**Conflict of Interest Disclosure**   The authors declare that they have no conflicts of interest.

## 2. Related Work

**Event-based Object Detection.** Most event-based object detection methods use ANN approaches, such as RED (Perot et al., 2020), ASTMNet (Li et al., 2022), and RVT (Gehrig & Scaramuzza, 2023), which demonstrate impressive detection performance but come with high energy consumption. To improve efficiency, several works explore SNN-based detection with event datasets. For example, Hybrid-SNN (Kugele et al., 2021) combines an SNN backbone with an ANN head. Spiking-DenseNet (Li et al., 2022) is notable for applying SNNs to event-based object detection using the SSD architecture. Multi-scale structures have also been introduced to event-based detection, including spiked-based feature pyramids (Zhang et al., 2023) and spike-based fusion modules (Fan et al., 2024). Current SNN-based object detectors can be broadly categorized into two types. Converted models such as Spiking-YOLOv4 (Wang et al., 2023b) facilitate fast and accurate object detection from event streams but require a large number of time steps to match original ANN performance. Directly trained SNNs, such as EMS-YOLO (Su et al., 2023), surpass ANN-to-SNN conversion methods, requiring only a few time steps for real-time inference. Recent studies incorporating more bio-brain-inspired elements into SNN training (Mao et al., 2025; Li et al., 2025), consistently achieve lower energy consumption. This trend indicates that increasing biological plausibility can improve the energy efficiency of SNNs. Motivated by this insight, we pursue a more biologically grounded design across three levels: neurons, spike encoding, and network architectures.

**Spiking Neural Network Architectures.** Residual architectures are central to enabling deep SNNs. Prior efforts such as SEW-ResNet (Fang et al., 2021a) and MS-ResNet (Hu et al., 2024) extended SNNs' depth beyond 100 layers. However, SEW-ResNet fundamentally relies on integer multiplication operations within residual blocks, while MS-ResNet focuses exclusively on spiking residual paths while neglecting non-spiking structures in shortcut connections. For object detection tasks with variable dimensions and channel numbers, these non-spiking convolutions incur significant energy overhead. EMS-ResNet (Su et al., 2023) introduces a fully spiking residual network architecture to fully exploit the energy efficiency intrinsic to SNNs. In this work, we retain the effective residual downsampling design of EMS-ResNet to preserve essential floating-point operations in shortcut connections. Moreover, most SNNs applied feedforward design, which prevents intermediate layers from perceiving multi-scale information during training. In our work, we introduce a bidirectional multi-scale

architecture that overcomes the limitations of prior unidirectional backbones. Beyond network structure, enhancing neurons also boosts SNN performance. Examples like PLIF (Fang et al., 2021b) demonstrate this (it learns better membrane timing to improve spiking network learning).

## 3. Method

### 3.1. Noise Filtering Neuron Architecture

Biological vision neurons possess intrinsic noise-filtering capabilities (Xi et al., 2024). Studies (Zang & Marder, 2023) indicate that populations of lateral pyloric neurons exhibit highly noise-resistant properties, demonstrating remarkable stability. Neurons in SNNs are typically highly simplified abstractions of biological neurons. They approximate the complex spatial integration process of the dendritic layer as a single, instantaneous, linear summation operation (Delorme et al., 1999), and often neglect the biological mechanisms such as inhibitory synapses, and build-in noise-processing capabilities. Notably, certain inhibitory synapses can lower the overall excitability of the postsynaptic neuron, making it more difficult for weak and asynchronous noise inputs to trigger spikes and thereby enabling natural noise suppression. Studies such as (Otopalik et al., 2017; Zang & Marder, 2023) have revealed that the noise resistance properties of neuronal populations are insensitive to the spatial location of input signals. Furthermore, the brain can encode, transmit, and integrate diverse information across multiple temporal scales flexibly (Yan et al., 2025). In the field of computer signal processing, studies (Zhao et al., 2019) have confirmed that noise can be filtered out using soft-thresholding techniques. Based on the aforementioned principles regarding the spatiotemporal perception of information by neurons, we propose the Spatio-Temporal Adaptive Threshold Noise-Filtering Neuron (STATNFN) architecture, an enhanced version of the base neuron that adaptively filters membrane potential inputs, enabling effective suppression of noise and redundant information while reducing spike rates. Spatio-Temporally Adaptive Threshold Noise Filtering (Figure 2 (b)) operates on the input membrane potential $X^{T \times B \times C \times H \times W}$ and outputs the noise-filtered membrane potential $Y^{T \times B \times C \times H \times W}$.

(Duan et al., 2022) has demonstrated that spatiotemporal normalization can enhance SNN performance. First, we perform spatio-temporal normalization on $X$ as initialization for the spatio-temporal features of the neuronal population:

$$\hat{X}_{t,b,c,h,w} = (\gamma_c \frac{X_{t,b,c,h,w} - \mu_c}{\sigma_c + \epsilon} + \beta_c)\lambda_t, \qquad (1)$$

where $\gamma$, $\beta$, and the temporal modulation parameters $\lambda_t$ are learnable scaling factors.

Based on the spatial-location insensitivity demonstrated by neuronal populations in noise resistance, we compute spatial

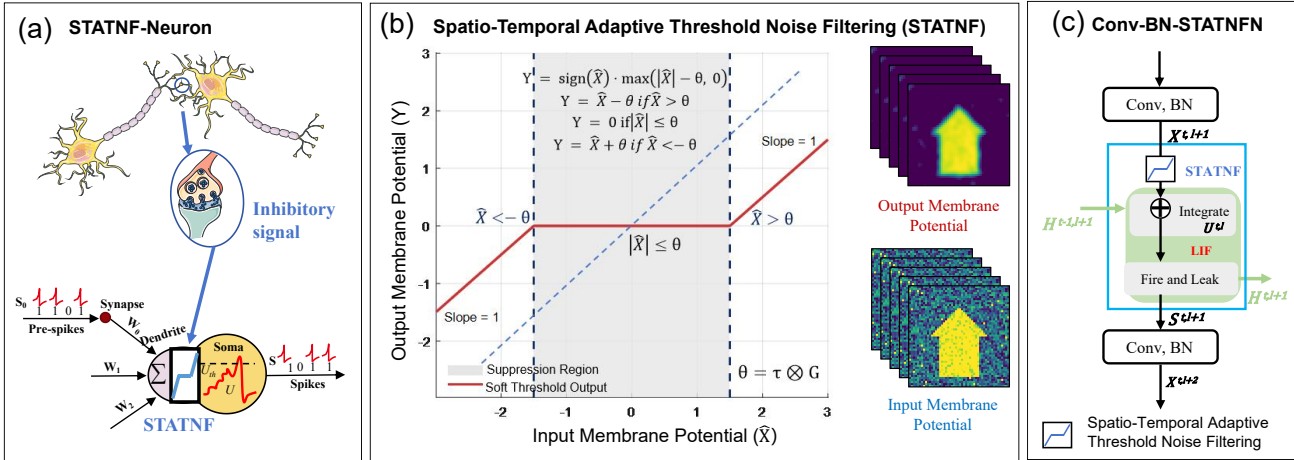

*Figure 2.* (a) Bio-Inspired Noise-Filtering Neuron: the Spatio-Temporal Adaptive Threshold Noise Filtering Neuron (STATNFN) architecture is inspired by inhibitory signals for noise filtering. (b) Spatio-Temporal Adaptive Threshold Noise Filtering: its core soft-threshold filtering technique eliminates membrane potential noise, reducing firing rates. (c) Trainable Spiking-Conv-STATNFN Block: STATNF integrates with fundamental neurons and deep learning modules for training.

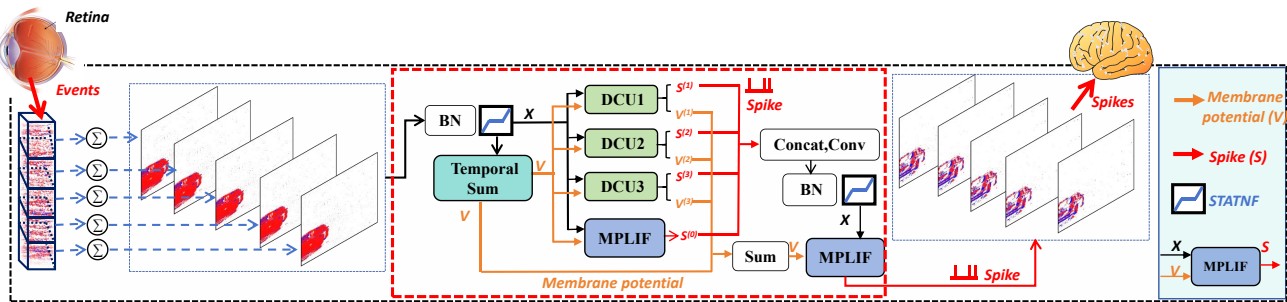

*Figure 3.* Retinal-Inspired Events-to-Spikes (E2S) framework. To simulate retinal motion perception, denoising, memory, and multi-scale perception, the framework utilizes the temporal information processing capabilities of the SNN network to perceive motion information, employs STATNFN neurons for denoising, designs MPLIF neurons with memory enhancement to strengthen the memory of the SNN, and incorporates Dilated Convolution Units (DCUs) with different dilation rates to perceive multi-scale information. E2S ensures that the input features to the SNN are spiking signals.

feature statistics through spatial averaging $\mathbf{G} \in \mathbb{R}^{T \times B \times C}$:

$$\mathbf{G} = \frac{1}{HW} \sum_{i=1}^{H} \sum_{j=1}^{W} |\hat{\mathbf{X}}|. \tag{2}$$

To mimic the bio-vision systems's capacity for cross-timescale integration, we apply a learnable temporal projection matrix $\mathbf{W} \in \mathbb{R}^{T \times T}$ to $\mathbf{G}$:

$$\tau = \sigma(\mathbf{W}\mathbf{G} + \mathbf{b}), \tag{3}$$

where $\mathbf{b} \in \mathbb{R}^T$ is a temporal bias vector, $\sigma$ denotes the sigmoid activation. Based on Eq. (2) and Eq. (3), we derive the spatio-temporally adaptive soft threshold $\Theta$ for noise filtering in STATNF:

$$\Theta = \tau \otimes \mathbf{G}, \tag{4}$$

where $\otimes$ denotes element-wise multiplication (Hadamard product). The final STATNF operation can be expressed as:

$$\mathbf{Y} = \begin{cases} \hat{\mathbf{X}} - \Theta & \text{if } \hat{\mathbf{X}} > \Theta \\ 0 & \text{if } |\hat{\mathbf{X}}| \leq \Theta \\ \hat{\mathbf{X}} + \Theta & \text{if } \hat{\mathbf{X}} < -\Theta \end{cases}. \tag{5}$$

STATNF can be integrated with various spiking neurons (Figure 2 (c)) as a feature preprocessing module (**Appendix A.3**). Compared to other threshold-modulated neuronal approaches, the proposed neuron architecture is more flexible and can be applied on top of common base neuron models, fundamentally addressing the impact of noise at the input level. The STATNFN architecture fully captures the spatiotemporal dynamics of biological neuronal noise processing described above (Otopalik et al., 2017; Yan et al., 2025), making it more biologically plausible.

### 3.2. Events-to-Spikes Representation

The rich synaptic activity in the retina provides a wide range of abilities for processing visual information, such as recognizing motion, filtering out noise, forming memories, and perceiving scale (Xi et al., 2024). To emulate these retinal functions while preserving the spiking characteristics of retinal-visual cortex connectivity, we propose the Events-to-Spikes Representation (E2S) framework (Figure 3).

(Mao et al., 2025) suggests that the temporal processing of

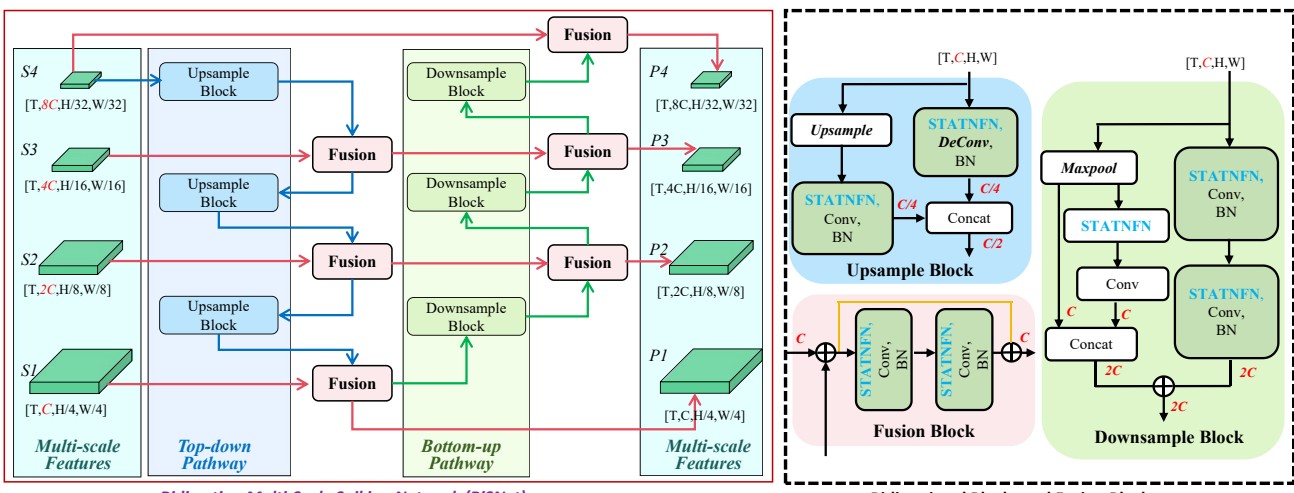

*Figure 4.* The overview architecture of Bidirectional Multi-Scale Spiking Network (BiSNet).

---

**Algorithm 1** Events-to-Spikes Representation (E2S)

**Require:** Input event tensor $X \in \mathbb{R}^{T \times B \times C \times H \times W}$
1: $X \leftarrow \text{STATNF}(\text{BN}(X))$
2: $\mathcal{V} \leftarrow \sum_{t=0}^{T-1} X_t$
3: $S_1 \leftarrow \text{MPLIF}(X, \mathcal{V})$
4: $\mathcal{V}_{\text{sum}} \leftarrow \mathcal{V}$
5: **for** each dilation rate $d \in \{1, 2, 3\}$ **do**
6:     **// DCU Start for dilation** $d$
7:     $S_1^{(d)} \leftarrow \text{STATNF+MPLIF}(X, \mathcal{V})$
8:     $F_{\text{depth}}^{(d)} \leftarrow \text{DepthwiseConv}_d(S_1^{(d)})$
9:     $\mathcal{V}_1^{(d)} \leftarrow \sum_{t=0}^{T-1} (F_{\text{depth}}^{(d)})_t$
10:    $S_2^{(d)} \leftarrow \text{STATNF+MPLIF}(F_{\text{depth}}^{(d)}, \mathcal{V}_1^{(d)})$
11:    $F_{\text{point}}^{(d)} \leftarrow \text{PointwiseConv}(S_2^{(d)})$
12:    $\mathcal{V}_2^{(d)} \leftarrow \sum_{t=0}^{T-1} (F_{\text{point}}^{(d)})_t$
13:    $S_3^{(d)} \leftarrow \text{STATNF+MPLIF}(F_{\text{point}}^{(d)}, \mathcal{V}_2^{(d)})$
14:    $\mathcal{V}_{\text{DCU}}^{(d)} \leftarrow \mathcal{V}_1^{(d)} + \mathcal{V}_2^{(d)}$
15:    **// DCU End**
16:    $\mathcal{V}_{\text{sum}} \leftarrow \mathcal{V}_{\text{sum}} + \mathcal{V}_{\text{DCU}}^{(d)}$
17: **end for**
18: $F_{\text{fuse}} \leftarrow \text{Conv}_{1 \times 1}\Big(\text{Concat}([S_1, S_3^{(1)}, S_3^{(2)}, S_3^{(3)}])\Big)$
19: $F_{\text{out}} \leftarrow \text{STATNF+MPLIF}(F_{\text{fuse}}, \mathcal{V}_{\text{sum}})$
    **return** $F_{\text{out}}$

---

SNNs can capture motion information, but it also leads to noise accumulation. E2S employs a STATNFN architecture to achieve noise filtering capabilities. Additionally, to enhance the memory capabilities of the E2S framework, we have developed Memory-enhanced PLIF (MPLIF) neurons by modifying the membrane potential integration mechanisms in PLIF, enabling robust long-term temporal retention. PLIF can be described as:

$$V_t = V_{t-1} + \frac{1}{\tau}(-(V_{t-1} - V_{rest}) + X_t), \quad (6)$$

$$S_t = H(V_t - V_{th}), \quad (7)$$

where $V_t$ denotes the membrane potential at timestep $t$. $X_t$ represents the input to neuron. $\tau$ is the learnable membrane time constant. $H(.)$ is the Heaviside step function. $V_{th}$ denotes the firing threshold. From Eq. (6), it can be seen that the PLIF neuron cannot perceive information from later time steps during earlier time steps, and as time progresses, earlier information will also be forgotten. (Fang et al., 2023) proposed Parallel Spiking Neurons, affirming the critical importance of both parallel processing and temporal awareness. To enable the membrane potential at each $t$ to perceive features across the entire temporal sequence $\mathcal{T} = \{1, 2, ..., T\}$, while enhancing the neuronal memory retention, we modify the initial membrane potential at time $t = 0$:

$$\mathcal{V} = \sum_{t=0}^{T-1} X_t. \quad (8)$$

The modified Memory-enhanced PLIF can be expressed as MPLIF($X$,$V$), where $X$ denotes the input features and $V$ represents the initial membrane potential with enhanced memory information, with $V$ being modifiable as required.

To enable the E2S framework to possess scale perception capabilities similar to the retina, we utilize three Dilated Convolution Units (DCUs) with different dilation rates to extract representations at various scales. In each DCU, the features sequentially pass through: (STATNF + MPLIF) → dilated Depthwise Conv → (STATNF + MPLIF) → Pointwise Conv → (STATNF + MPLIF). Finally, the features from multiple DCU branches are concatenated, and the final spike features are output through (STATNF + MPLIF). The complete process of E2S is illustrated in Algorithm 1.

### 3.3. Bidirectional Multi-Scale Spiking Network

Current SNNs face the challenge of weak multi-scale modeling capabilities. Information transmission between visual

cortices is not a simple unidirectional flow from lower-level (V1) to higher-level (V4) areas. As (Xin et al., 2025) systematically clarifies, feedback signals from higher-to-lower visual cortices play a central role in parsing complex visual scenes. Inspired by this bidirectional propagation mechanism, we propose the Bidirectional Multi-Scale Spiking Network (BiSNet) (Figure 4).

First, four spiking residual blocks produce feature maps at four scales: $S_1, S_2, S_3, S_4$, with strides of 4, 8, 16, and 32, respectively. These features $\{S_1, S_2, S_3, S_4\}$ are then refined through bidirectional information flow (top-down and bottom-up) with lateral connections. BiSNet ultimately outputs enhanced features $\{P_1, P_2, P_3, P_4\}$.

**Top-down Pathway:** This pathway propagates semantic information from high-level features to low-level features. Starting from the deepest feature $S_4$, we upsample and fuse it with the adjacent shallower feature $S_3$, and so on.

**Bottom-up Pathway:** It propagates spatial details from low-level features to high-level features. Starting from $P_1$, we downsample and fuse it with $P_2$, and so on.

**Fusion Block:** Fusion$(X, Y)$ combines input features through element-wise addition followed by a convolutional block. The block consists of two convolutional layers with batch normalization and spiking activation.

In BiSNet, all fundamental modules (Downsample Block, Upsample Block, Fusion Block) incorporate STATNF-Neuron design (Figure 4).

# 4. Experiments

## 4.1. Experimental Settings and Performance Metrics

**Datasets.** We conduct experiments on NCAR (Sironi et al., 2018) for event-based object recognition, and on two benchmarks for event-based object detection: N-Caltech101 (Orchard et al., 2015) and the Gen1 (De Tournemire et al., 2020). **Implementation Details.** Our models are trained using the AdamW optimizer (Kinga et al., 2015) on a single NVIDIA RTX 4090 GPU. NCAR Dataset: 30 epochs with batch size 64, initial learning rate $5 \times 10^{-3}$, and weight decay $1 \times 10^{-2}$. N-Caltech101 Dataset: 100 epochs with batch size 8, initial learning rate $1 \times 10^{-3}$, and weight decay $1 \times 10^{-4}$. GEN1 Dataset: 35 epochs with batch size 8, initial learning rate $1 \times 10^{-3}$, and weight decay $1 \times 10^{-4}$. Further implementation details are provided in **Appendix** A.4. **Performance Metrics.** Accuracy, and mAP$_{50:90}$, mAP$_{50}$ are used to evaluate the tasks of recognition and detection, respectively. The procedures for AC/MAC and energy calculations are detailed in the **Appendix** A.5.

## 4.2. Effective Tests

**Object Recognition.** We present in Table 1 the network types, data representations (Rep.), parameters, inference time steps (T), accuracy (Acc.), firing rate (FR), AC/MAC, and energy consumption for various methods. STATNF-BiSNet establishes a new state-of-the-art (SOTA) performance on NCAR with exceptional efficiency: (i) it achieves the highest accuracy (96.1%) among all compared methods; (ii) it maintains the lowest firing rate (0.09) and energy consumption (0.02 mJ) among SNNs.

**Object Detection.** STATNF-BiSNet achieves SOTA detection performance on the Gen1 and N-Caltech101 datasets. On the Gen1 dataset (Table 2), STATNF-BiSNet surpasses all SNN detectors with a mAP$_{50:90}$ of 0.4 and a mAP$_{50}$ of 0.691. Unlike other SNN methods, CREST (AAAI2025) and MSD (CVPR2025) also employ bio-brain-inspired mechanisms. These approaches primarily simulate the brain's response to multi-scale information, demonstrating significant improvements in performance and energy efficiency. This represents a 2.8% improvement in mAP$_{50}$ over the best-performing brain-inspired SNN, MSD, at an energy cost that is 4.3× lower. On the N-Caltech101 dataset (Table 3), STATNF-BiSNet sets new benchmarks with a mAP$_{50:90}$ of 45.0% and a mAP$_{50}$ of 63.5%. The visualization of E2S and the detection results are shown in Figure 5.

## 4.3. Ablation Studies

**Component Effectiveness Validation.** The ablation study in Table 4 highlights the effectiveness of three components: STATNF, E2S, and Bidirectional Multi-Scale (BiS). The use of STATNF significantly reduces the FR , with FR decreasing from 0.371 to 0.26, while also improving accuracy (mAP increases from 0.373 to 0.4). Incorporating E2S further decreases the firing rate, leading to improved accuracy (FR: 0.26 to 0.133, mAP: 0.4 to 0.414). Although the use of BiS increases the number of parameters (from 8.9M to 10M), it improves the model's accuracy (mAP: 0.414 to 0.45). Notably, with the application of the three bio-vision-inspired modules proposed in this paper, the parameter count increases only slightly (+1.1M), while the accuracy sees a significant improvement (mAP +7.7%).

**Ablation Study of STATNFN.** To further validate the effectiveness of the proposed STATNFN architecture, we conducted ablation studies on recognition and detection tasks. In most cases, STATNFN can greatly reduce the network's FR while maintaining or even improving task accuracy. As shown in Figure 6, ablation experiments are performed across 30 combinations of 6 network architectures (SEW ResNet (Fang et al., 2021a), MS ResNet (Hu et al., 2024), EMS ResNet (Su et al., 2023)) and 5 neurons (LIF(Delorme et al., 1999), PLIF (Fang et al., 2021b), PSN (Fang et al., 2023), SPSN3 and SPSN5(Fang et al., 2023)).

*Table 1.* Comparison with sate-of-the-art models on the NCAR dataset.

| Method | Type | Rep. | Params | T | Acc. | Firing Rate | AC/MAC (G) | Energy (mJ) |
|---|---|---|---|---|---|---|---|---|
| YOLOE (Cannici et al., 2019) | ANN | Leaky Surface | - | - | 0.927 | 1 | 0.16 | 0.75 |
| Asynet (Messikommer et al., 2020) | ANN | VoxelGrid | - | - | 0.944 | 0.067 | 0.16 | 0.05 |
| EvS-S (Li et al., 2021) | GNN | Graph | - | - | 0.931 | 1 | - | - |
| Gabor-SNN (Sironi et al., 2018) | SNN | HAT | - | 5 | 0.789 | - | - | - |
| HybridSNN (Kugele et al., 2021) | SNN | HIST | - | 5 | 0.770 | - | - | - |
| Squeeze-1.1 (Cordone et al., 2022) | SNN | VoxelCube | 0.72M | 5 | 0.846 | 0.251 | 0.02 | 0.36 |
| Mobile-64 (Cordone et al., 2022) | SNN | VoxelCube | 18.81M | 5 | 0.917 | 0.171 | 4.2 | 83.34 |
| Dense121-24 (Cordone et al., 2022) | SNN | VoxelCube | 3.93M | 5 | 0.904 | 0.336 | 2.25 | 37.76 |
| VGG-11 (Cordone et al., 2022) | SNN | VoxelCube | 9.23M | 5 | 0.924 | 0.120 | 0.61 | 12.68 |
| Dense121-16 (Fan et al., 2024) | SNN | VoxelCube | 1.76M | 5 | 0.937 | 0.147 | 0.06 | 1.22 |
| CREST$_{CSPdarknet}$ (Mao et al., 2025) | SNN | MESTOR | 3.41M | 5 | 0.949 | 0.165 | 0.05 | 0.04 |
| CREST$_{DenseNet121-16}$ (Mao et al., 2025) | SNN | MESTOR | 1.95M | 5 | 0.952 | 0.146 | 0.07 | 0.04 |
| STATNF-BiSNet (ours) | SNN | E2S | 6.1M | 5 | **0.961** | **0.09** | **0.02** | **0.02** |

*Table 2.* Comparison with sate-of-the-art models on the Gen1 dataset.

| Method | Type | Rep. | Head | Params | T | mAP$_{50:90}$ | mAP$_{50}$ | FR | AC/MAC (G) | Energy(mJ) |
|---|---|---|---|---|---|---|---|---|---|---|
| EGO-12 (Zubić et al., 2023) | ANN | EGO-12 | YOLOv6 | 140M | - | 0.504 | - | 1 | 84.34 | 387.96 |
| S-Center (Bodden et al., 2024) | ANN | HIST | CenterNet | 12.97M | - | 0.278 | - | 1 | 6.13 | 28.21 |
| HsVT (Xu et al., 2025) | ANN+SNN | VoxelGrid | YOLOX | 17.2M | - | 0.478 | - | - | - | 68.84 |
| VC-Dense (Cordone et al., 2022) | SNN | VoxelCube | SSD | 8.20M | 5 | 0.189 | - | 0.372 | 2.33 | - |
| VC-Mobile (Cordone et al., 2022) | SNN | VoxelCube | SSD | 24.26M | 5 | 0.147 | - | 0.294 | 4.34 | - |
| LT-SNN (Hasssan et al., 2023) | SNN | HIST | YOLOv2 | 86.82M | 5 | 0.298 | - | - | 19.53 | - |
| EMS-10 (Su et al., 2023) | SNN | HIST | YOLOv3 | 6.20M | 5 | 0.267 | 0.547 | 0.211 | 5.90 | - |
| EMS-18 (Su et al., 2023) | SNN | HIST | YOLOv3 | 9.34M | 5 | 0.286 | 0.565 | 0.201 | 9.70 | - |
| EMS-34 (Su et al., 2023) | SNN | HIST | YOLOv3 | 14.40M | 5 | 0.310 | 0.590 | 0.178 | 32.99 | - |
| S-Center (Bodden et al., 2024) | SNN | HIST | CenterNet | 12.97M | 5 | 0.229 | - | 0.174 | 6.38 | - |
| TR-YOLO (Yuan et al., 2024) | SNN | HIST | YOLOv3 | 8.70M | - | - | 0.451 | - | - | - |
| SFOD (Fan et al., 2024) | SNN | VoxelCube | SSD | 14.40M | 5 | 0.321 | 0.593 | 0.240 | 6.72 | 7.26 |
| EAS-SNN (Wang et al., 2024) | SNN | ARSNN | YOLOX | 8.92M | 3 | 0.354 | 0.675 | - | 8.08 | 15.9 |
| SpikeYOLO (Luo et al., 2024) | SNN | HIST | YOLOv8 | 23.10M | 5 | 0.389 | 0.664 | - | - | 19.7 |
| CREST (Mao et al., 2025) | SNN | MESTOR | YOLOv4 | 7.61M | 5 | 0.360 | 0.632 | 0.167 | 8.39 | 6.31 |
| MSD (Li et al., 2025) | SNN | HIST | SSD | 7.80M | 5 | 0.389 | 0.663 | - | - | 6.5 |
| STATNF-BiSNet (ours) | SNN | E2S | SSD | 24.19M | 5 | **0.400** | **0.691** | **0.132** | **1.65** | **1.48** |

**Note:** T denotes the number of time steps. FR represents the firing rate. AC/MAC refers to the computational cost. The values reported in the table are referenced from the literature (Mao et al., 2025). The gray background indicates that the method employs a bio-inspired SNN mechanism. The best results are highlighted in **bold**.

*Table 3.* Comparison with SOTA models on the N-Caltech101 dataset.

| Method | Type | Rep. | Params | mAP$_{50}$ |
|---|---|---|---|---|
| YOLOE (Cannici et al., 2019) | ANN | Surface | - | 0.398 |
| AEGNN (Schaefer et al., 2022) | GNN | Graph | 20M | 0.595 |
| Asynet (Messikommer et al., 2020) | ANN | VoxelGrid | - | 0.615 |
| EvS-S (Li et al., 2021) | GNN | Graph | 0.9M | 0.346 |
| EAS-SNN (Wang et al., 2024) | SNN | ARSNN | 8.9M | 0.538 |
| STATNF-BiSNet (ours) | SNN | E2S | 10M | **0.635** |

*Table 4.* The contribution of each component to our method on the N-Caltech101 dataset.

| | Base | STATNF | E2S | BiS | Params | FR | mAP |
|---|---|---|---|---|---|---|---|
| (a) | ✓ | | | | 8.9M | 0.371 | 0.373 |
| (b) | ✓ | ✓ | | | 8.9M | 0.260 | 0.400 |
| (c) | ✓ | ✓ | ✓ | | 8.9M | 0.133 | 0.414 |
| (d) | ✓ | | | ✓ | 10M | 0.407 | 0.394 |
| (e) | ✓ | | ✓ | ✓ | 10M | 0.280 | 0.400 |
| (f) | ✓ | ✓ | | ✓ | 10M | 0.208 | 0.429 |
| ours | ✓ | ✓ | ✓ | ✓ | 10M | **0.141** | **0.450** |

Using STATNFN consistently reduces the network's firing rate, and over two-thirds of the methods show accuracy improvements to varying extents after using STATNFN. As shown in Table 5, for the detection task, different network architectures (MS ResNet (Hu et al., 2024), EMS ResNet (Su et al., 2023), *ours*) show that after using the STATNFN architecture, the model parameter counts remain nearly unchanged, while both the firing rate and energy consumption decrease significantly, and accuracy improves to varying extents. The energy consumption analysis of the STATNF module is provided in **Appendix** A.6. An analysis of the proposed method's robustness to noise is provided in **Appendix** A.7. The ablation experiments of the components (DCU, MPLIF, STATNF) in E2S are provided in

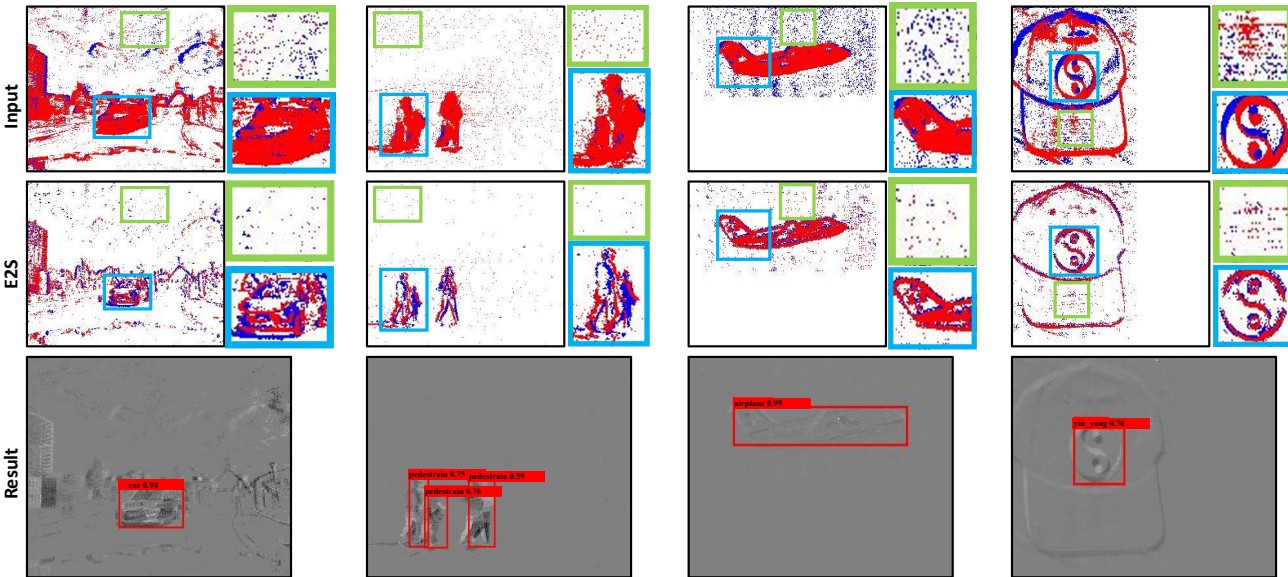

*Figure 5.* Visualizations of the Event-to-Spike Representation (E2S) and final results on the Gen1 and N-Caltech101 datasets. Green box area: E2S significantly reduces noise in raw event data. Blue box area: E2S retains key information with robustness.

*Table 5.* Performance comparison of different network structures and neuron types on the N-Caltech101 dataset. The baseline (BASE) uses the SPSN5 neuron (Fang et al., 2023), while STATNFN denotes the integrated STATNF-SPSN5 neuron.

| Network | Neuron | Params | FR (%) | AC/MAC (G) | Energy (mJ) | mAP$_{50:90}$ (%) | mAP$_{50}$ (%) |
|---|---|---|---|---|---|---|---|
| MS-ResNet | BASE | 14.8M | 19.3 | 0.171 | 0.151 | 37.8 | 53.9 |
| (Hu et al., 2024) | STATNFN | 14.8M | 14.9$_{-4.4}$ | 0.132$_{-0.039}$ | 0.119$_{-0.032}$ | 38.0$_{+0.2}$ | 54.3$_{+0.4}$ |
| EMS-ResNet | BASE | 14.7M | 30.0 | 1.31 | 1.18 | 37.0 | 54.6 |
| (Su et al., 2023) | STATNFN | 14.7M | 16.0$_{-14.0}$ | 0.699$_{-0.611}$ | 0.630$_{-0.550}$ | 37.9$_{+0.9}$ | 54.9$_{+0.3}$ |
| STATNF-BiSNet | BASE | 10.0M | 28.0 | 0.564 | 0.507 | 40.0 | 58.7 |
| (ours) | STATNFN | 10.0M | **14.1**$_{-13.9}$ | 0.284$_{-0.280}$ | 0.255$_{-0.252}$ | **45.0**$_{+5.0}$ | **63.5**$_{+4.8}$ |

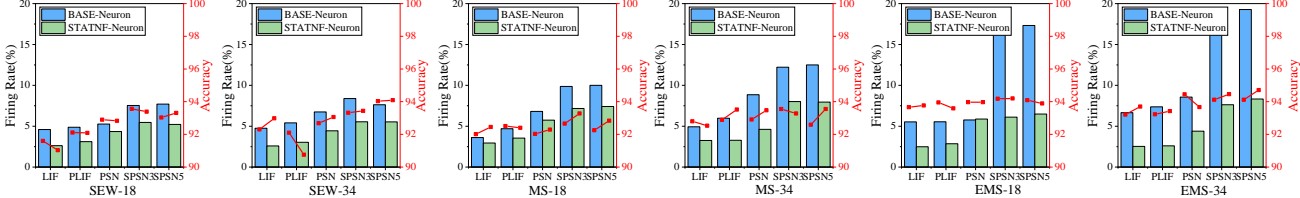

*Figure 6.* Impact of STATNFN on spike firing rates and accuracy across different SNN architectures and neuron configurations (NCAR).

**Appendix** A.8. Additional analysis of the tables and figures is provided in **Appendix** A.9, A.10, A.11.

## 5. Conclusion

Inspired by visual systems, we develop a bio-vision-inspired Event-SNN framework that better fits event data through coordinated neuron-level and macro-structural designs. The STATNFN neuron introduces adaptive spatio-temporal noise filtering, and this effect is further amplified at the macro level through integration into the E2S encoder and the BiS-Net. This forms a multi-level noise-suppression pipeline that effectively filters redundant signals and reduces firing activity. Moreover, with MPLIF-based temporal memory and multi dilated convolutions in E2S, and bio-inspired bidi-

rectional pathways in BiSNet, the approach achieves robust multi-scale spatio-temporal integration and more reliable feature preservation. Extensive experiments on event data validate the approach's effectiveness and efficiency. As a successful bio-vision-inspired design, our framework offers insights with promising extensibility to future neuromorphic models and tasks. **Limitations and Future Work**: Like most works in this field (Fan et al., 2024; Wang et al., 2024; Luo et al., 2024; Xu et al., 2025; Li et al., 2025; Mao et al., 2025), this SNN model is currently not deployed on neuromorphic hardware due to resource limitations. The proposed components can be considered for testing real energy consumption on neuromorphic chips in the future. The bio-visually plausible Event-SNN architecture can be applied to broader tasks like object tracking, denoising, and depth

estimation. For multi-camera SLAM under extreme lighting or high velocities, combining our energy-efficient event representations with modern geometric solvers is a practical next step. Specifically, Guan et al. (Guan et al., 2026) derived the first analytical solutions for generalized relative pose estimation using two affine correspondences and analyzed the associated degeneracy cases. By addressing key theoretical bottlenecks in weak geometric configurations, their approach demonstrates high precision and computational efficiency. Incorporating their solvers into our robust Event-SNN framework can further improve multi-camera navigation in complex scenarios.

## Impact Statement

This paper presents work whose goal is to advance the field of Machine Learning. There are many potential societal consequences of our work, none which we feel must be specifically highlighted here.

## Acknowledgements

Supported by Beijing Science and Technology Plan (Z241100004224011); Shenzhen Science and Technology Program (KQTD20240729102051063); the National Natural Science Foundation of China (62425101,62332002); Guangdong Special Support Program for Young Top-notch Talents (2024TQ08X327).

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

# A. Appendix

### A.1. The Basics of Spike Neural Networks

Spike neural networks (SNNs) are a type of artificial neural network inspired by the biological brain's functioning. They communicate through discrete, asynchronous spikes, which allows for efficient and biologically plausible information processing. SNNs have the advantage of handling temporal information naturally, making them well-suited for processing event data. Compared to traditional frame-based ANNs, SNNs offer the potential for improved performance in tasks involving spatio-temporal processing. Additionally, SNNs have the advantage of low energy consumption compared to ANNs, making them highly promising in energy-constrained devices.

#### A.1.1. CONV-NEURON-BASED SNNS

Conv-Neuron-based SNNs can be constructed for network training **(Figure 2 (c) in the paper)**. A Conv-Neuron-based SNN layer consists of convolutional and Neuron components. The information propagation in this architecture can be described as follows:

$$U^{t,l} = H^{t-1,l} + X^{t,l}. \tag{9}$$

$$S^{t,l} = Hea(U^{t,l} - u_{th}). \tag{10}$$

$$H^{t,l} = V_{rest}S^{t,l} + (\beta U^{t,l}) \odot (1 - S^{t,l}). \tag{11}$$

Where $t$ and $l$ respectively represent timestamps and network layers. $V$ represents the membrane potential generated by integrating spatial dimension input $X^{t,l}$ and temporal dimension input $H^{t-1,l}$. $u_{th}$ is the threshold that determines whether the output spike tensor $S^{t,l}$ should be fired or kept as zero, with $Hea()$ representing the Heaviside step function. $H^{t,l}$ represents the internal state of neurons propagating over time, where $\beta = e^{\frac{-dt}{\tau}}$ reflects the decay factor, and $\odot$ denotes element-wise multiplication. This paper focuses on constructing networks based on Conv-based SNNs.

#### A.1.2. LEARNING METHODS IN SPIKE NEURAL NETWORKS

As traditional backpropagation algorithms cannot be directly applied to SNNs due to the non-differentiability of spike signals, alternative learning rules have been developed. One prominent example is Spike-Time-Dependent Plasticity (STDP) (Toyoizumi et al., 2004), a biologically inspired unsupervised learning mechanism that modifies the synaptic strengths between neurons based on the precise timing of pre-synaptic and post-synaptic spikes. The most effective and commonly used learning method currently is supervised learning based on surrogate gradient (Neftci et al., 2019; Shrestha & Orchard, 2018; Zenke & Ganguli, 2018).

The training method used in this paper is based on surrogate gradients, where the surrogate gradient function employed can be described as:

$$g(x) = \frac{1}{\pi} \arctan(\frac{\pi}{2}\alpha x) + \frac{1}{2}. \tag{12}$$

During the backpropagation stage of the network, the gradient is given by:

$$g'(x) = \frac{\alpha}{2(1 + (\frac{\pi}{2}\alpha x)^2)}. \tag{13}$$

In this paper, $\alpha = 2.0$, $V_{rest} = 1.0$, and $V_{th} = 1.0$.

### A.2. Event Camera and Event Representation

The output of an event camera with a resolution of $H \times W$ can be represented as an event sequence, denoted as $E = {e_i}_1^N$, where $e_i = (x_i, y_i, t_i, p_i)$. Here, $p_i \in (-1, 1)$ represents the polarity of a brightness change that occurs at time $t_i$ and pixel position $(x_i, y_i)$. The change is triggered for the pixel $(x_n, y_n)$ at timestamp $t_n$ when the log-intensity $\ln L$ changes beyond the pre-defined threshold $\theta$. This dynamic visual sensing mechanism is depicted by the inequality:

$$\ln L(x_n, y_n, t_n) - \ln L(x_n, y_n, t_n - \Delta t_n) \geqslant p_n\theta, \tag{14}$$

where the polarity $p_n \in \{1, -1\}$ indicates whether the brightness is increasing or decreasing, and $\Delta t_n$ represents the temporal sampling interval of DVS at a pixel.

Voxel grid is a classic event data representation method based on accumulation. Specifically, for each set of input events defined by $\{(x_i, y_i, t_i, p_i)\}_{i=0,\dots,N-1}$, we divide them into $B$ distinct bins that span various time instances. For every individual time step ($t$), the construction of the corresponding event volume proceeds in the following manner:

$$t_i^* = \frac{(B-1)(t_i - t_1)}{t_N - t_1},$$

$$V(x, y, t) = \sum_i p_i k(x - x_i) k(y - y_i) k(t - t_i), \tag{15}$$

$$k(a) = \max(0, 1 - |a|),$$

where $k(a)$ corresponds to the bilinear sampling kernel. To better extract temporal features from event data, we treat the time dimension as an input channel. Extracting features from channels in the network is equivalent to extracting temporal information.

### A.3. The Complete Mathematical Definition of STATNFN

The most significant distinction between the STATNFN neuronal architecture and prior neuron improvement approaches is its ability to integrate with any base neuron without modifying the internal structure or neuronal dynamics formulation. STATNFN draws inspiration from the spatiotemporal perception mechanism of biological neuronal populations in response to noise, which enhances the sensitivity of neuronal spiking to the input information $X$.

The classical neuronal formulation (e.g., PLIF) is expressed as follows:

$$S_t = H(V_t - V_{th}), \tag{16}$$

$$V_t = V_{t-1} + \frac{1}{\tau}(-(V_{t-1} - V_{rest}) + X_t). \tag{17}$$

Most neuronal modification strategies (such as PLF and ALIF) focus on refining parameters like $\tau$ or $V_{th}$, while overlooking the significance of membrane potential input $X_t$.

STATNFN can be regarded as an optimization of the input $X_t$. This approach maintains the advantages of the base neuron while enhancing its ability to filter out redundant features and noise. Taking the PLIF neuron as an example, we outline the specific computational flow of the STATNFN architecture:

$$S_t = H(V_t - V_{th}), \quad (Output\ spike) \tag{18}$$

$$V_t = V_{t-1} + \frac{1}{\tau}(-(V_{t-1} - V_{rest}) + Y), \quad (Membrane\ potential) \tag{19}$$

$$\mathbf{Y} = \begin{cases} \hat{\mathbf{X}} - \Theta & \text{if } \hat{\mathbf{X}} > \Theta \\ 0 & \text{if } |\hat{\mathbf{X}}| \le \Theta \\ \hat{\mathbf{X}} + \Theta & \text{if } \hat{\mathbf{X}} < -\Theta \end{cases}, \quad (Soft - threshold\ filtering) \tag{20}$$

$$\Theta = \tau \otimes \mathbf{G}, \quad (Spatiotemporal\ soft - threshold) \tag{21}$$

$$\tau = \sigma(\mathbf{WG} + \mathbf{b}), \quad (Temporal - aware\ of\ the\ input) \tag{22}$$

$$\mathbf{G} = \frac{1}{HW} \sum_{i=1}^{H} \sum_{j=1}^{W} |\hat{\mathbf{X}}|, \quad (Spatial - aware\ of\ the\ input) \tag{23}$$

$$\hat{X} = (\gamma_c \frac{X_t - \mu_c}{\sigma_c + \epsilon} + \beta_c)\lambda_t, \quad (Normalization) \tag{24}$$

where $V_t$ denotes the membrane potential after neuronal dynamics at timestep $t$. $X_t$ represents the input to neuron. $\tau$ is the learnable membrane time constant. $H(.)$ is the Heaviside step function. $V_{th}$ denotes the firing threshold.

The noise in input data (particularly in Event data) tends to be discrete and subtle. However, traditional neurons may amplify the impact of noise through temporal accumulation. STATNF can learn a spatiotemporal threshold to filter out such noise. The concept of soft-threshold noise filtering was proposed earlier in the field of computer signal processing. We introduce this technique into the improvement of spiking neurons. The theoretical foundations of soft-threshold noise filtering can be referenced in literature (Zhao et al., 2019), which will not be reiterated in this paper.

Taking the LIF neuron as an example, we conducted simulation tests and analysis on the soft-threshold filtering technique. As shown in Figure 7, it can be observed that adding noise to the input data significantly increases the firing rate (FR), while soft-threshold filtering can effectively mitigate the impact of noise. In this paper, the soft threshold used is an optimal value learned through spatiotemporal features. Since no parameter learning was involved in the simulation tests, we used a fixed hyperparameter of 0.4, with the time step set to 0.001. The input at each time step was set to 1.3, with noise introduced as random values between 0 and 0.5 at each step.

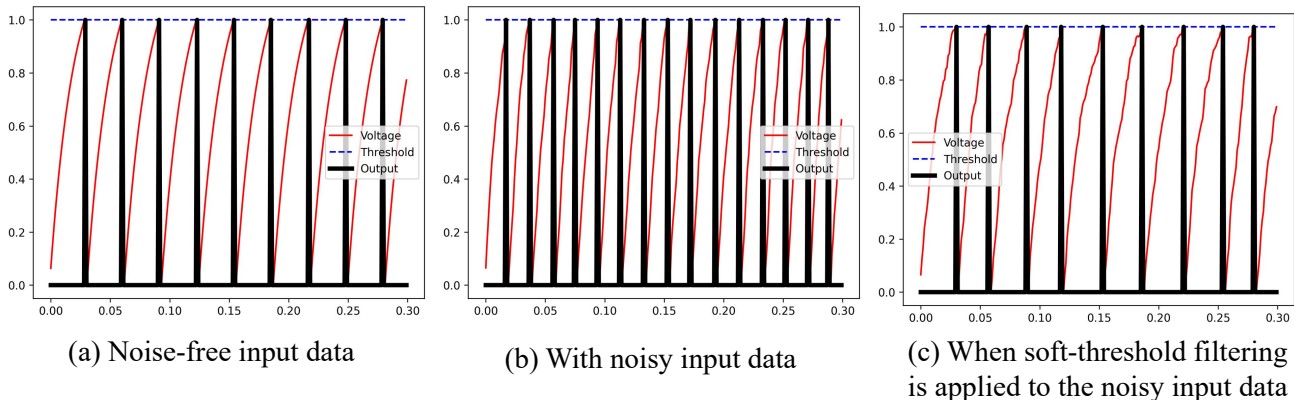

(a) Noise-free input data  (b) With noisy input data  (c) When soft-threshold filtering is applied to the noisy input data

*Figure 7.* Taking the LIF neuron as an example: (a) spike firing under ideal conditions (9 spikes), (b) spike firing with noisy input (15 spikes), and (c) spike firing with noisy input but using threshold filtering (9 spikes).

### A.4. More Implementation Details

For the NCAR dataset, spatial dimensional consistency across samples is achieved through nearest-neighbor interpolation, with all inputs resized to a standardized 64×64 resolution and temporal duration fixed at 100 ms. For the GEN1 dataset, temporal alignment is maintained by truncating each annotated bounding box to a 100 ms interval, while preserving its native 304×240 pixel spatial resolution throughout processing.

For all network components outside the E2S module, we implement STATNFN using STATNF+SPSN5 neurons. [C, 2C, 4C, 8C] represents the number of channels in the model, while K represents the number of Bidirectional Multi-Scale (BiS) (**Figure 4 in the paper**). These two parameters affect the scale of the model. For the NCAR and N-Caltech101 datasets, the model achieves optimal performance with K=2 and C=32. On the Gen1 dataset, the best results are achieved with K=2 and C=64. We conducted experiments on the NCAR dataset for the model-scale parameters (C and K) reflecting architectural

*Table 6.* Ablation study of BiS number (K) and channel width (C) in BiSNet on NCAR Dataset.

| Method | K | C | Params | FR | AC/MAC | Energy(mJ) | Acc. |
|--------|---|---|--------|-----|--------|-----------|------|
| (a) | 1 | 32 | **3.3 M** | 0.114 | **0.012** | **0.011** | 0.951 |
| (b) | 1 | 64 | 13.3M | 0.089 | 0.034 | 0.030 | 0.957 |
| (c) | 2 | 32 | 6.1M | **0.077** | 0.013 | 0.012 | **0.961** |

dimensions, as shown in Table 6.

## A.5. Energy Consumption

The energy consumption of SNNs in neuromorphic hardware are usually assessed based on the number of computational operations (Su et al., 2023). In ANNs, each operation involves floating-point multiplications and additions (MAC). SNNs exhibit high energy efficiency in neuromorphic hardware because only neurons involved in spike generation contribute to accumulation calculations (AC), and computations can be performed with roughly the same number of synaptic operations. However, many current SNNs introduce additional MAC operations due to their design flaws. Hence, we quantify the energy consumption of the original SNN as $E_{SNN} = \sum E_l$, where the energy of the $l$-th layer can be calculated as:

$$E_l = T \times (S_{fr} \times E_{AC} \times OP_{AC}) + E_{MAC} \times OP_{MAC}, \tag{25}$$

where $T$ represents the time step, $S_{fr}$ denotes the firing rate (FR), and $OP_{AC}$ and $OP_{MAC}$ represent the numbers of AC and MAC operations, respectively. The energy cost for per 32-bit floating point AC/MAC operation is 0.9/4.6 pJ (Horowitz, 2014). Eq. 25 shows that a low firing rate $S_{fr}$ directly reduces the energy consumption of SNNs. This is one key reason why our design includes noise filtering to reduce the firing rate, thus lowering energy consumption. STATNF-BiSNet's energy consumption (BiSNet's energy consumption plus STATNF's energy cost).

## A.6. Energy Analysis of STATNF

STATNF is applied throughout the entire network, and its energy consumption is also significant. STATNF reduces the firing rate (FR) of SNNs, thereby decreasing the overall energy consumption of the SNN. However, STATNF itself incurs an energy cost (Equations 1-5), yet this additional energy expenditure is substantially outweighed by the overall energy savings it enables.

STATNF's energy consumption primarily comprises the operations and costs specified in Equations 1-5: Eq. 1 requires per element one subtraction, one division, one multiplication, and one addition; Eq. 2 involves computing the average of all elements within the pooling kernel for each output element (AC)); Eq. 3 involves matrix multiplication (MAC) and bias addition (AC)); Eq. 4-5 requires element-wise multiplication and a comparator. We calculated the energy consumption of SNNs both with and without STATNF on the N-CAR dataset, as detailed below:

*Table 7.* Energy consumption analysis using STATNF (ACC. 96.1%)

| Component | FR (%) | AC/MAC (G) | Energy (mJ) |
|---|---|---|---|
| STATNF | - | 0.0007 | 0.0021 |
| BiSNet | - | 0.0159 | 0.0143 |
| STATNF-BiSNet (our) | 9.25 | 0.0166 | 0.0164 (Total) |

*Table 8.* Energy consumption analysis without using STATNF (ACC. 94.75%)

| Component | FR (%) | AC/MAC (G) | Energy (mJ) |
|---|---|---|---|
| BiSNet | 39.88 | 0.0686 | 0.0667 |

Conclusion: Although the STATNF mechanism introduces a modest increase in energy consumption, it significantly reduces the overall energy cost by filtering redundant features and lowering the spike firing rate. As a result, employing STATNF yields an energy advantage of 4.1× (0.0667/0.0164).

## A.7. Noise Robustness Analysis

To validate the noise robustness of the proposed neuron, we conducted experiments comparing the proposed STATNFN neuron with the base neuron N (SPSN5) on noise-injected datasets. First, different noise ratios were added to the original event data (Figure 8, top). We then evaluated two trained detection models: the STATNFN-based model and the baseline model without STATNFN.

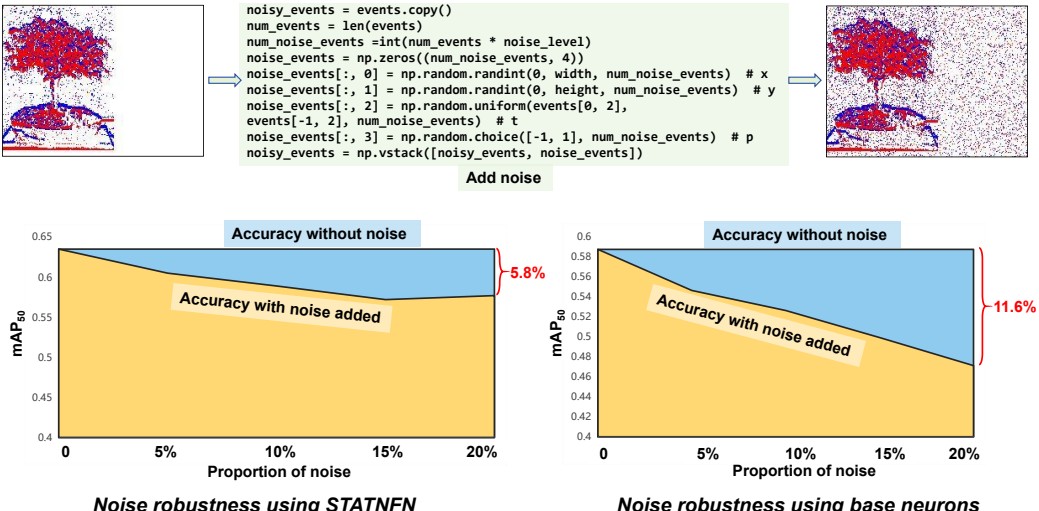

*Figure 8.* STATNFN noise robustness analysis.

The STATNFN model achieved 63.5% $mAP_{50}$. After adding 5%, 10%, 15% and 20% noise, its $mAP_{50}$ values were 60.5%, 58.9%, 57.2% and 55.7% respectively. The baseline model started at 58.7% $mAP_{50}$, decreasing to 54.6%, 52.6%, 49.9% and 47.1% under corresponding noise conditions (Figure 8, bottom).

Using post-noise mAP drop as the robustness metric, our method showed only 5.8% degradation at 20% noise versus the base neuron's 11.6% reduction. This demonstrates our approach significantly enhances noise robustness.

### A.8. Ablation Experiments of the Components in E2S

We conduct ablation experiments on the components DCU, MPLIF, and STATNF in E2S using the N-CAR datasetTable 9.

*Table 9.* Ablation experiments of the components in E2S.

|  | DCU | MPLF | STATNF | Accuracy (%) |
|---|---|---|---|---|
| (a) |  | ✓ | ✓ | 95.7 |
| (b) | ✓ |  | ✓ | 95.7 |
| (c) | ✓ | ✓ |  | 94.8 |
| E2S (ours) | ✓ | ✓ | ✓ | **96.1** |

### A.9. Bidirectional Architecture Superiority

To further validate the effectiveness of our proposed bidirectional SNN architecture, we conducted comparative experiments with the classical unidirectional SNN structure (EMS-ResNet) on two datasets: NCAR and N-Caltech101.

As shown in Table 10, for the NCAR classification task, our approach achieves higher accuracy (96.1% vs. 95.9%) with fewer parameters (6.1M vs. 10.7M), lower computational cost (0.013 G vs. 0.034 G), reduced spike firing rate (0.077 vs. 0.087), and lower energy consumption (0.012 mJ vs. 0.031 mJ).

Similarly for the N-Caltech101 detection task (Table 11), our method demonstrates superior performance with fewer parameters (10.0M vs. 14.7M), higher accuracy (42.9% vs. 37.9%), reduced computational cost (0.398G vs. 0.690G), lower spike firing rate (0.160 vs. 0.208), and decreased energy consumption (0.358mJ vs. 0.630mJ).

In summary, the proposed architecture outperforms state-of-the-art unidirectional networks in both accuracy and energy efficiency across multiple metrics.

*Table 10.* Performance comparison of unidirectional and bidirectional architectures on NCAR classification task.

| Method | Params | FR | AC/MAC(G) | Energy(mJ) | Acc. |
|---|---|---|---|---|---|
| EMS (Only down) | 10.7 M | 0.087 | 0.034 | 0.031 | 0.959 |
| Our (Bidirectional) | **6.1M** | **0.077** | **0.013** | **0.012** | **0.961** |

*Table 11.* Performance comparison of unidirectional and bidirectional architectures on N-Caltech101 detection task.

| Method | Params | FR | AC/MAC(G) | Energy(mJ) | mAP$_{50:90}$ |
|---|---|---|---|---|---|
| EMS (Only down) | 14.7M | 0.160 | 0.699 | 0.630 | 0.379 |
| Our (Bidirectional) | **10.0M** | 0.208 | **0.398** | **0.358** | **0.429** |

## A.10. More Result Analysis

**Table 1 in the paper** presents the network types (Type), representations (Rep.), parameter counts (params), accuracy (Acc.), spike firing rates, operations (AC/MAC), and energy consumption for each method. Among these representations, Leaky Surface, VoxelGrid, HAT, and HIST can all be considered accumulation-based methods. Only our approach ensures that the data input to the SNN is in spike form, maintaining the biological and rational integrity of the entire model. Although MESTOR also utilizes SNNs, its final output remains in floating-point values rather than in binary format (0 and 1). Our method benefits from noise-filtering neurons and the E2S structure, resulting in a spike firing rate lower than that of other SNN methods; our method achieves a spike firing rate of only 8%, while other methods have rates exceeding 14%. Our approach demonstrates optimal accuracy and the lowest energy consumption, achieving a good balance between accuracy and energy efficiency.

**Table 2 in the paper** provides a comparison with other methods used for event-based object detection, where the accuracy evaluation metrics employed are $mAP_{50:90}$ and $mAP_{50}$. Common detection heads for event-based object detection include the YOLO series, SSD, and CenterNet. Among the YOLO series, the method with the highest accuracy is SpikeYOLO ($mAP_{50:90}$ = 38.9%, $mAP_{50}$ = 66.4%). EAS-SNN proposes an RNN-based representation method for event data, but it increases computational complexity and cannot guarantee that the features input to the SNN are in binary (0, 1) format. SFOD primarily innovates through spike feature fusion. None of these methods specifically address the network's ability to handle noise and control spike firing rates, nor do they adequately consider the rationality of event data representation. Our method addresses these challenges while focusing on controlling spike firing rates, ensuring the rationality of event data representation, and incorporating bidirectional multi-scale fusion. Although the current best ANN methods achieve higher accuracy, they also have significantly higher energy consumption. Our method is 262× more energy-efficient than the best ANN method (EGO-12). The energy consumption efficiency of our method is is 262× more energy-efficient than the ANN-SNN hybrid structure HsVT.

**Table 3 in the paper** presents a comparison with other methods on the N-Caltech101 dataset. There are few SNN methods available for this dataset, so we only compared with the EAS-SNN method. The ARSNN data representation proposed by this method can significantly improve detection accuracy, but its overall performance still falls short of our approach. Our method achieves an 11.2% improvement in $mAP_{50:90}$ while maintaining a comparable number of parameters.

**Figure 5 in the paper** primarily validates the effectiveness of E2S through visual visualization. The magnification of the green box shows that after using E2S, discrete noise can be reduced, while the magnification of the blue box demonstrates that key features can be preserved after applying E2S, making the data more effective.

From **Figure 6 in the paper**, it can be seen that the detection accuracy of SPSN5 + EMS-ResNet34 reaches its optimal level, which is why we chose SPSN5 as the base neuron for our model. To further illustrate the impact of using STATNF on network accuracy and spike firing rate, we present the scatter plot below (Figure 9).

## A.11. More Visualization of Detection Results

Figure 10 presents additional visualization results on the N-Caltech101 and GEN1 datasets. The top two rows show detection results on N-Caltech101, while the bottom two rows display detection results on GEN1.

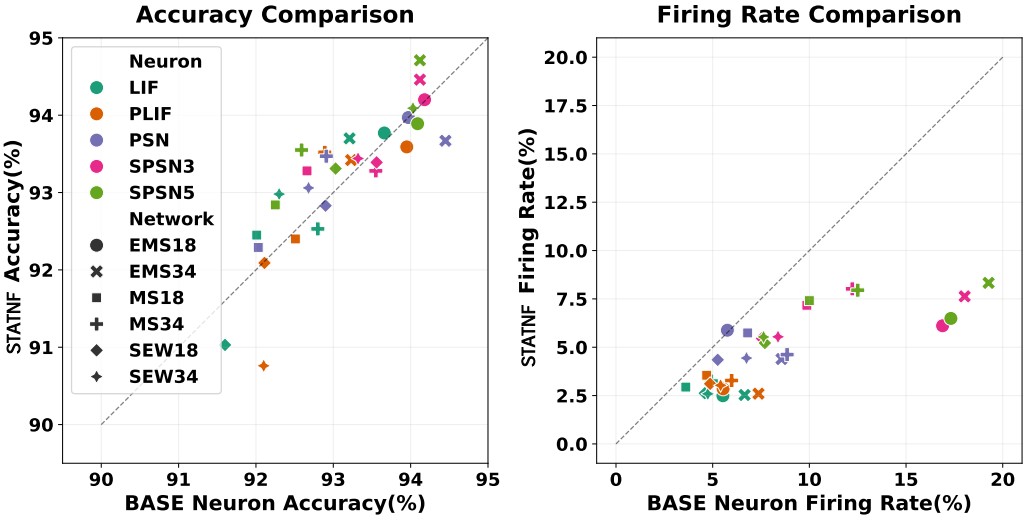

*Figure 9.* Impact of STATNFN on spike firing rates and accuracy.

*Figure 10.* More visualization of detection results.

