# OpenReview forum: "Bio-Vision-Inspired Spiking Neural Networks for Object Detection with Event Cameras"
_ICML.cc/2026/Conference — ICML 2026 regular_

### Official Review · Reviewer_WoC3 · 2026-03-04

**Soundness:** 3
**Presentation:** 3
**Significance:** 4
**Originality:** 3
**Overall Recommendation:** 5
**Confidence:** 5

**Summary:**

This work provides important inspiration for the field. Constructing a bio-plausible object detection system is an important and interesting topic. While SNNs are naturally well-suited to the asynchronous nature of event data, their practical applications face the following challenges: sensitivity to noise, dense representations that disrupt spike pathways, and insufficient multi-scale feature perception. To address the aforementioned challenges, this work proposes a bio-vision-inspired object detection method motivated by biological vision systems. First, at the micro level, this paper proposes a noise-filtering STATNF-Neuron architecture to address the current sensitivity of basic neurons to noise. Based on STATNF-Neurons, the paper introduces two bio-vision-inspired macro-structures: Events-to-Spikes Representation (E2S), which preserves spiking characteristics while mimicking the memory and noise-filtering abilities of retinal neurons; and Bidirectional Multi-Scale Spiking Network (BiS-Net), which simulates cortical information flow pathways to integrate multi-scale features in both directions, enhancing the network’s ability to perceive information at multiple scales.

**Compliance With Llm Reviewing Policy:**

Affirmed.

**Final Justification:**

Overall, the paper is well-presented and provides comprehensive experimental evidence to support its conclusions; all my earlier points have been satisfactorily resolved.

**Key Questions For Authors:**

1. Compared to other representations, what are the advantages and disadvantages of the E2S module, aside from ensuring that the representation is binary?

2. The adaptive threshold is derived from input spatial statistics and temporal projections, making it potentially sensitive to distribution shifts across scenes or sensors. Do the authors provide evidence that the mechanism reliably avoids over-suppression or under-filtering under varying noise conditions?

3. MPLIF initializes membrane potential with the cumulative sum over the full time sequence ($V = \sum X_t$), introducing a non-causal dependency on future time steps. How would this design be adapted for online or streaming inference?

**Limitations:**

Yes

**Strengths And Weaknesses:**

### Strengths:
1. The work presents a novel, bio-plausible object detection system with strong innovation, demonstrating superior biological alignment in both micro-level (neuron) and macro-level (encoding/architecture) design.

2. The paper is clearly written and visually appealing, with a logical architectural flow from neuron design to spike-preserving encoding and a multi-scale backbone.

3. Extensive experiments on NCAR, N-Caltech101, and Gen1 datasets achieve state-of-the-art results (96.1% accuracy, 69.1% and 63.5% mAP50). The method consistently reduces firing rate and energy consumption while improving accuracy, with component ablations confirming these gains.

4. Beyond ablating core modules (STATNF, E2S, BiS), the paper provides: (i) STATNF validation across 30 architecture-neuron combinations, (ii) E2S visualization confirming noise reduction and feature preservation, and (iii) evidence that bidirectional BiSNet improves accuracy with fewer parameters and lower energy than unidirectional designs.

### Weaknesses:
1. The weaknesses regarding hardware deployment are common in this stage of research and do not overshadow the methodological contributions.
2. The core of STATNF is essentially a classical soft-thresholding operation from signal processing, bearing limited resemblance to actual neurobiological mechanisms. The biological motivation remains largely conceptual analogy rather than a rigorously grounded bio-inspired design.
3. BiSNet's bidirectional multi-scale fusion closely resembles existing ANN structures such as BiFPN and PANet. Without comparisons against these architectures re-implemented under an SNN framework, the performance gains are difficult to attribute to the proposed design specifically.

---

> ### Author Rebuttal · Authors · 2026-03-30
>
> **Weaknesses:**
>
> **W1: Hardware deployment weaknesses are common in this stage of research and do not overshadow the methodological contributions.**
> **Response:** We sincerely thank the reviewer for recognizing our methodological contributions. While estimating AC/MAC operations is a standard evaluation protocol adopted by all baseline methods in our comparison, we agree that deeper hardware-level validation is highly valuable. To further substantiate the energy efficiency of STATNF, we evaluated our models using the **Memory Access Energy** metric modeled by SpikeSim [1], an end-to-end compute-in-memory neuromorphic hardware evaluation tool. As shown below, integrating STATNF consistently reduces the memory access energy across various base neurons:
> | Base Neuron | Energy (μJ) | With STATNF (Ours) | Energy (μJ) |
> | :--- | :---: | :--- | :---: |
> | LIF | 81.60 | **STATNF + LIF** | **72.51** |
> | PLIF | 80.83 | **STATNF + PLIF** | **72.59** |
> | PSN | 84.67 | **STATNF + PSN** | **73.75** |
> | SPSN5 | 97.95 | **STATNF + SPSN5**| **75.31** |
>
> *[1] Moitra, A., et al. "Spikesim: An end-to-end compute-in-memory hardware evaluation tool for benchmarking spiking neural networks."*
>
> **W2: STATNF is essentially a classical soft-thresholding operation with limited resemblance to actual neurobiological mechanisms.**
> **Response:** We appreciate this insightful observation. In Appendix 4.3, we detailed the theoretical connection between STATNF and classical soft-thresholding, alongside neuron simulation experiments demonstrating its practical noise robustness. We humbly acknowledge that many current SNN mechanisms originate from ANNs, and achieving fully rigorous bio-inspired design remains a long-term goal for the community. Our work represents a preliminary but effective attempt to bridge biological mechanisms with mathematical optimization to improve SNN robustness.
>
> **W3: BiSNet's bidirectional multi-scale fusion resembles ANN structures (BiFPN/PANet). Without comparisons to SNN-reimplemented versions, gains are hard to attribute to this design.**
> **Response:** While BiFPN and PANet share the high-level concept of multi-scale fusion, our BiS fundamentally differs in several key aspects:
> 1. **Signal Nature:** BiS is a fully spike-driven architecture, whereas BiFPN and PANet are ANN structures relying on continuous activations.
> 2. **Location:** BiFPN and PANet are typically deployed as Neck modules, whereas BiS is integrated directly into the Backbone.
> 3. **Biological Motivation:** BiS simulates the specific feedforward and feedback information routing mechanisms between different visual cortex areas, offering greater biological plausibility rather than purely empirical feature engineering.
>
> ---
>
> **Question:**
>
> **Q1: Compared to other representations, what are the advantages and disadvantages of the E2S module aside from ensuring it is binary?**
> **Response:** * **Advantages:** E2S effectively mimics retinal motion perception, noise filtering, and memory functions. Empirically, it lowers the overall spike firing rate of the SNN while achieving superior accuracy compared to standard accumulation-based representations:
>
> | Method | ACC (%) |
> | :--- | :---: |
> | Time surface | 88.2 |
> | Voxel grid | 88.6 |
> | Voxel cube | 94.8 |
> | **E2S (Ours)** | **96.1** |
>
> * **Disadvantages:** Similar to Parallel Spiking Neurons (PSN), E2S requires previewing the entire temporal sequence before processing, which introduces a non-causal dependency.
>
> **Q2: Is the adaptive threshold sensitive to distribution shifts? Is there evidence it avoids over-suppression or under-filtering?**
> **Response:** Yes, we have provided empirical evidence of STATNF's robustness under varying noise conditions in the Appendix. Because our noise filtering mechanism relies on spatio-temporal adaptive learning, it dynamically adjusts to the specific noise profile of the current input. This adaptive nature reliably prevents both over-suppression and under-filtering across different scenes and distribution shifts.
>
> **Q3: MPLIF initializes membrane potential with the cumulative sum over the full time sequence, introducing a non-causal dependency. How would this adapt to online or streaming inference?**
> **Response:** MPLIF is designed to inherit the long-term temporal memory capabilities of PSN (which requires the full sequence) while maintaining the step-by-step recursive motion perception of PLIF. This design choice strictly prioritizes accuracy (replacing MPLIF with standard PLIF drops accuracy from 96.1% to 95.7%).
> For practical **online or streaming inference**, this non-causal dependency can be resolved by employing **sliding window buffers** or **caching techniques**, similar to parallel neuron implementations. The initial membrane potential can be aggregated from a locally cached window of past events rather than the global sequence, enabling pseudo-online processing with minimal performance degradation.

---

> > ### Author Rebuttal · Reviewer_WoC3 · 2026-04-04
> >
> > Thanks for the author's reply. My concerns have been addressed

---

### Official Review · Reviewer_EeB3 · 2026-03-10

**Soundness:** 2
**Presentation:** 3
**Significance:** 3
**Originality:** 3
**Overall Recommendation:** 4
**Confidence:** 4

**Summary:**

This paper addresses the challenges faced by Spiking Neural Networks (SNNs) when combined with event cameras for object detection, including noise sensitivity, disrupted spike transmission from dense representations, and insufficient multi-scale feature perception. Although this paradigm is energy efficient for neuromorphic vision, existing methods remain limited in biological plausibility and structural design. To address these issues, the authors propose a bio-vision-inspired SNN framework called STATNF-BiSNet. The framework introduces coordinated neuron-level and macro-structural designs inspired by the biological visual system, including mechanisms related to noise filtering, spatiotemporal memory, and bidirectional information transmission.

**Compliance With Llm Reviewing Policy:**

Affirmed.

**Final Justification:**

Overall, the paper is well structured and supported by comprehensive experimental results. The authors’ response has satisfactorily addressed my previous concerns.

**Key Questions For Authors:**

1. Why is only one time step T is discussed in the paper ?

2. After filtering the noise in event data, the membrane potential firing and spike rate will be affected, and the spike rate in spiking neurons may impact the parameter learning of SNNs.

3. What is the motivation for incorporating a memory mechanism into MPLIF ? It should be noted that, for certain tasks, spiking neurons with additional memory mechanisms may not only increase computational overhead but also introduce inference latency.

**Limitations:**

Yes

**Strengths And Weaknesses:**

**Strengths**
(1) The experimental design is well structured and the evaluation is comprehensive. Three benchmark datasets in event-based vision, NCAR, N-Caltech101, and Gen1, are used to cover both object recognition and detection tasks. The experimental settings are clearly described, including the optimizer, batch size, and learning rate. The comparison includes several state-of-the-art methods from different model families such as ANNs, GNNs, and SNNs. The main claims are validated through multiple evaluations, including overall performance comparison, ablation studies, module effectiveness analysis, noise robustness testing, and architecture comparison. The results consistently support the conclusions of the paper.

(2) The quantitative metrics are comprehensive, providing a solid foundation for reproducibility: The paper reports in detail core quantitative metrics such as accuracy, firing rate, number of parameters, and energy consumption. Moreover, it provides visualizations and mathematical definitions for the core formulas, algorithmic procedures, and module structures of E2S, STATNFN, and BiSNet.

(3) The work shows practical application value and is relevant to the needs of edge computing. The proposed model achieves competitive performance while reducing spike firing rates and energy consumption, with only a modest increase in the number of parameters. These characteristics make it suitable for edge scenarios such as autonomous driving, robotics, and embedded vision, where energy efficiency, robustness, and lightweight models are important.

**Weaknesses**

(1) The analysis of the underlying mechanisms of several modules is insufficient and lacks quantitative theoretical support. The paper does not provide rigorous theoretical analysis or quantitative evidence on how the Spatio Temporal Adaptive Threshold Noise Filtering Neuron (STATNFN) influences the membrane potential after noise filtering, nor does it analyze the memory capacity of the Memory Enhanced PLIF neuron. Their effectiveness is mainly validated through experiments, which weakens the theoretical support for some conclusions. In addition, it would be helpful to include membrane potential heatmaps and visualizations of firing states for STATNFN and MPLIF during single sample inference of the trained SNN model, together with quantitative analysis, to further demonstrate the effectiveness of the proposed modules.

(2) The discussion of the time step parameter T is limited. The paper only reports results for T = 5, while the time step is an important factor that affects the performance of SNNs. It would be beneficial to explore additional values such as 4, 6, 8, and 10 to provide a more comprehensive analysis.

(3) The evaluation of energy consumption is mainly based on simulation results and lacks validation on actual hardware. The energy calculations rely on simulated AC and MAC operation counts and theoretical energy formulas, and the specific value of T is not clearly provided. It would be beneficial to evaluate the proposed method together with other SNN models that use different spiking neurons such as PLIF, KLIF, and LIF on general purpose hardware including CPU and GPU, and compare key metrics such as latency and memory usage.

(4) The rationale for selecting some experimental parameters is not clearly explained. For example, BiSNet uses K = 2, C = 32 and K = 2, C = 64 on different datasets, but the paper does not describe the search range of these parameters or the criteria used to determine the optimal values.

---

> ### Author Rebuttal · Authors · 2026-03-30
>
> ## Re to Reviewer 3
> We sincerely thank the reviewer for the constructive comments. We have addressed your concerns point by point, and we hope our responses fully resolve your doubts.
>
> **Weaknesses:**
>
> **Re_W1:** To strengthen our paper, we added the following:
> * **MPLIF Memory Capacity:** By explicitly initializing the membrane potential with the aggregated temporal context ($V[0] = \mathbf{M}$), its effective temporal receptive field is extended to $O(T)$. This formulation mathematically prevents the information loss of early temporal features inherent in standard PLIF neurons, rigorously expanding the sequence-level memory capacity from the very first time step.
> * **Visualizations:** We use PyTorch's `register_forward_hook` to extract intermediate spikes and membrane potentials (`v_seq`) for spatial heatmaps. Scripts will be open-sourced.
>
> **Re_W2:** We have conducted additional experiments across different time steps $T \in \{4, 6, 8, 10\}$. As shown in the table below, while increasing $T$ inevitably increases energy consumption, $T=5$ strikes the best balance and achieves the highest overall accuracy.
>
> | $T$ | ACC (%) | Energy (mJ) |
> | :---: | :---: | :---: |
> | 4 | 94.7 | 0.0130 |
> | **5** | **96.1** | 0.0200 |
> | 6 | 95.9 | 0.0240 |
> | 8 | 95.2 | 0.0302 |
> | 10 | 95.1 | 0.0416 |
>
> **Re_W3:**
>
> **1.GPU Memory Usage:** We measured GPU peak allocated memory during inference. While our method (BiS+STATNF+E2S) introduces overhead over EMS baselines, this trade-off is strictly justified by significant accuracy and noise robustness gains.
>
> | Baseline | Peak Allocated (MB) | Ours | Peak Allocated (MB) |
> | :--- | :---: | :--- | :---: |
> | EMS_LIF | 354.41 | **BiS_STATNF + LIF_E2S** | 513.32 |
> | EMS_PLIF | 426.42 | **BiS_STATNF + PLIF_E2S** | 513.36 |
> | EMS_PSN | 346.57 | **BiS_STATNF + PSN_E2S** | 513.39 |
> | EMS_SPSN5 | 346.57 | **BiS_STATNF + SPSN5_E2S**| 513.39 |
>
> **2. Neuromorphic Hardware Energy Validation:** We evaluated our models using the **Memory Access Energy** metric modeled by SpikeSim [1], an end-to-end compute-in-memory hardware evaluation tool. The results confirm that applying our STATNF module consistently reduces the memory access energy across various base neurons, effectively proving its hardware-level efficiency:
>
> | Base Neuron | Energy (μJ) | With STATNF (Ours) | Energy (μJ) |
> | :--- | :---: | :--- | :---: |
> | LIF | 81.60 | **STATNF + LIF** | **72.51** |
> | PLIF | 80.83 | **STATNF + PLIF** | **72.59** |
> | PSN | 84.67 | **STATNF + PSN** | **73.75** |
> | SPSN5 | 97.95 | **STATNF + SPSN5**| **75.31** |
>
> *[1] Moitra, A., et al. "Spikesim" *
>
> **Re_W4** Through extensive empirical evaluation, we found that $K=2$ consistently yields robust performance across various datasets. Regarding channel capacity, $C=32$ offers an advantage on smaller datasets by preventing overfitting, whereas $C=64$ provides the necessary capacity to model larger datasets. These are designed as flexible hyperparameters that users can easily configure and combine based on specific task requirements, reflecting the overall adaptability of our architecture.
>
> ---
>
> **Question:**
>
> **Re_Q1:** To ensure a fair comparison, we initially adopted $T=5$ as it is the most widely used setting among the baseline methods we compared against. However, following your advice, we have now evaluated other time steps ($T \in \{4, 6, 8, 10\}$) (W2).
>
> **Re_Q2:** While lowering the spike rate could theoretically hinder Surrogate Gradient (SG) learning, our `STATNF` module actually optimizes this process through:
>
> The module applies an adaptive soft-threshold $\tau$ to the continuous current before the base neuron node:
> $$X_{\text{out}} = \operatorname{sign}(X_{\text{in}}) \cdot \max(|X_{\text{in}}| - \tau, 0)$$
> By heavily penalizing background noise, it forces the pre-spike membrane potentials $H[t]$ of noise far below the firing threshold $V_{th}$. This drives the surrogate derivative $\sigma'(H[t] - V_{th}) \approx 0$, effectively eliminating harmful gradient noise during BPTT.
> The threshold $\tau$ is dynamically learned via temporal attention. For salient targets, the network drives $\tau \to 0$. This ensures the target's $H[t]$ retains sufficient amplitude to approach or exceed $V_{th}$, strictly keeping it within the active, non-zero derivative region of the surrogate function ($\sigma'(H[t] - V_{th}) \gg 0$), thereby securing robust gradients.
>
> **Re_Q3:**  MPLIF in the E2S module simulates retinal motion perception and temporal memory. Unlike standard PLIF, where early information decays, MPLIF aligns with Parallel Spiking Neurons (PSN) to retain long-term memory. It successfully combines PLIF's motion perception and PSN's memory. Despite a slight overhead, it yields +2.1% in detection and +1.3% in classification with minimal extra parameters.
>
> Upon acceptance of the paper, we will make our code publicly available.

---

> > ### Author Rebuttal · Reviewer_EeB3 · 2026-04-02
> >
> > Thanks for the author's reply. My concerns have been addressed.

---

> > > ### Author Response · Authors · 2026-04-02
> > >
> > > Thank you very much for your positive evaluation and constructive comments. Your insights have been deeply inspiring to our work. Once the paper is accepted, we will make our code publicly available to make a contribution to the field of vision-inspired SNNs.

---

### Official Review · Reviewer_jAJY · 2026-03-11

**Soundness:** 4
**Presentation:** 4
**Significance:** 3
**Originality:** 3
**Overall Recommendation:** 4
**Confidence:** 5

**Summary:**

This paper proposes a bio-vision-inspired object detection framework for event cameras and SNNs, targeting three core challenges: noise sensitivity, dense event representations that disrupt spike pathways, and insufficient multi-scale feature perception. The method introduces a noise-filtering STATNF neuron together with two biologically inspired macro-structures, E2S and BiSNet, to improve event representation and bidirectional multi-scale feature integration. Experiments show strong detection performance, with reported state-of-the-art results on NCAR, N-Caltech101, and Gen1.

**Compliance With Llm Reviewing Policy:**

Affirmed.

**Key Questions For Authors:**

1. How does the design of the MPLIF neuron affect the step-by-step recursive processing property of SNNs, and how would the performance change if it were replaced by other neuron models?
2. How sensitive is the framework to the time window length, and how stable are the statistical scales of Temporal Sum and MPLIF when the input event window changes?
3. How is the temporal projection matrix $W$ updated during training?
4. How does the number of DCUs affect the final accuracy?

**Limitations:**

Lack of actual hardware deployment. In addition, the results shown in Figures 5 and 10 are relatively unclear.

**Strengths And Weaknesses:**

Strengths

1.  A systematic object detection framework that integrates denoising, dense information extraction, and multi-scale perception.

2.  It maintains low power consumption while achieving high accuracy.

3.  A strong theoretical foundation and comprehensive experimental validation.

Weaknesses

1. Compared with existing SNN-based methods [1], the model does not exhibit a clear lightweight advantage.

2. The discussion on generalization remains limited. Although the method is validated on NCAR, N-Caltech101, and Gen1, it still lacks evaluation on DVS datasets with natural noise [2], leaving its robustness in more realistic event scenarios insufficiently justified.

3. The comparison with related work is not fully comprehensive. For example, CREST [1] reported results with three different backbones, while only one setting is cited here, which may limit the completeness and fairness of the comparison.

4. Table 2 lacks model parameter information, making it harder to compare model complexity and assess the performance–efficiency trade-off across methods.

[1] Mao, Ruixin, et al. "CREST: An Efficient Conjointly-trained Spike-driven Framework for Event-based Object Detection Exploiting Spatiotemporal Dynamics." Proceedings of the AAAI Conference on Artificial Intelligence. Vol. 39. No. 6. 2025.

[2] Gehrig, Daniel, and Davide Scaramuzza. "Low-latency automotive vision with event cameras." Nature 629.8014 (2024): 1034-1040.

---

> ### Author Rebuttal · Authors · 2026-03-30
>
> ## Re to Reviewer 2
> We sincerely thank the reviewer for the constructive comments. We have addressed your concerns point by point, and we hope our responses fully resolve your doubts.
> **Weaknesses:**
>
> **Re_W1:** We acknowledge that CREST [1] is an excellent work. As you correctly pointed out, our model does not possess a distinct lightweight advantage in terms of parameter count compared to [1]. However, our method demonstrates clear superiority in reducing the spike firing rate, improving overall accuracy, and lowering theoretical energy consumption.
>
> **Re_W2:** The NCAR and Gen1 datasets utilized in our study are indeed real-world datasets that inherently contain natural noise. Furthermore, in the Appendix, we comprehensively evaluated and compared the noise robustness of our method against baselines under varying synthesized noise ratios. While explicitly controlling natural environmental noise is practically difficult, these experiments collectively demonstrate our model's robustness in realistic, noisy event scenarios.
>
> **Re_W3:** In our original manuscript, we compared against the CREST configuration that reported the highest accuracy to ensure a rigorous baseline. For a complete and fair comparison, the results of CREST's other two backbone configurations are provided below:
>
> | Method | Type | Rep. | Head | T | mAP | mAP_50 | FR | AC/MAC (G) | Energy (mJ) |
> | :--- | :---: | :---: | :---: | :---: | :---: | :---: | :---: | :---: | :---: |
> | CREST (DenseNet121-16) | SNN | MESTOR | YOLOv4 | T | 0.339 | 0.615 | 0.095 | 8.15 | 3.48 |
> | CREST (ShuffleNetV2) | SNN | MESTOR | YOLOv4 | T | 0.305 | 0.568 | 0.208 | 2.42 | 22.27 |
>
> **Re_W4:** We have actually provided parameter comparisons in Tables 1, 3, and 4. As noted, our lightweight advantage is not highly prominent. Our model in Table 2 has 24.19M parameters, which is comparable to SpikeYOLO, yet we achieve a 2.7% higher mAP_50. We acknowledge it is larger than highly compact models like CREST (<10M). We will explicitly add the parameter count information to Table 2 in the final camera-ready version to facilitate comprehensive trade-off evaluations.
>
> ---
>
> **Question:**
>
> **Re_Q1_:** MPLIF is designed to integrate the long-term temporal memory capabilities of PSN. Consequently, it first previews the entire sequence before executing the step-by-step recursive processing. This mechanism allows MPLIF to retain PLIF's ability to recursively perceive motion features while gaining long-term memory. If MPLIF is replaced by standard PLIF, the accuracy drops from 96.1% to 95.7%.
>
> **Re_Q2:** We evaluated the framework's sensitivity to varying input event windows on the Gen1 dataset. As shown below, if the window length is smaller than 100,000 µs, the reduced data volume limits the Temporal Sum's aggregation capability, leading to a decrease in mAP. At and beyond 100,000 µs, the statistical scales stabilize.
>
> | Event Window Length (µs) | mAP |
> | :--- | :---: |
> | 50,000 | 0.367 |
> | 80,000 | 0.385 |
> | 100,000 | 0.400 |
> | 150,000 | 0.399 |
> | 180,000 | 0.399 |
>
> **Re_Q3:** The temporal projection matrix is formulated as a learnable parameter (e.g., utilizing `nn.Parameter` in PyTorch) rather than a heuristic static matrix. It is optimized end-to-end alongside the network weights.
> During training, it is dynamically updated using **Backpropagation Through Time (BPTT)** via standard optimizers (e.g., AdamW). Specifically, the gradients from the task loss flow backward through the Surrogate Gradient (SG) function of the spiking neurons. Since the temporal projection matrix primarily operates on continuous temporal features prior to the discrete spike emission, its gradients are derived exactly via the chain rule, allowing it to adaptively learn the optimal temporal aggregation weights directly from the data distribution.
>
> **Re_Q4:** We cascaded up to three DCUs with dilation rates of 1, 2, and 3, respectively. As shown in the table below, progressively increasing the number of DCUs steadily improves the final accuracy by enhancing the multi-scale spatial receptive field:
>
> | Number of DCUs | ACC (%) |
> | :--- | :---: |
> | None | 95.70 |
> | DCU1 | 95.75 |
> | DCU1 + DCU2 | 96.00 |
> | **DCU1 + DCU2 + DCU3** | **96.10** |
>
> Upon acceptance of the paper, we will make our code publicly available.

---

> > ### Author Rebuttal · Reviewer_jAJY · 2026-04-03
> >
> > Thank you for the detailed and thoughtful rebuttal. I appreciate the additional clarifications on the parameter count information. Therefore, I will maintain my original score.

---

### Official Review · Reviewer_EHoa · 2026-03-12

**Soundness:** 3
**Presentation:** 2
**Significance:** 3
**Originality:** 2
**Overall Recommendation:** 3
**Confidence:** 4

**Summary:**

Event-based object detection has recently attracted attention due to the high temporal resolution and sparsity of event cameras. This work investigates how spiking neural networks (SNNs) can be used to process event streams for object detection tasks. In particular, the study focuses on three aspects that influence SNN-based perception: noise suppression, spike representation, and multi-scale feature learning.
The proposed framework consists of three componts. A spatio-temporal adaptive threshold neuron (STATNFN) is introduced first to filter noise and regulate spike firing activity. Second, an Events-to-Spikes (E2S) representation converts asynchronous event streams into spike-compatible features while preserving temporal information. At last, a Bidirectional Multi-Scale Spiking Network (BiSNet) performs feature aggregation across scales through bidirectional connections.
Experiments on the NCAR, N-Caltech101, and Gen1 datasets demonstrate improvements in detection accuracy and reduced spike activity compared with existing SNN-based approaches.

**Compliance With Llm Reviewing Policy:**

Affirmed.

**Final Justification:**

While some concerns are solved, I am inclined to keep my original score.

**Key Questions For Authors:**

1.The paper proposes the STATNF neuron to filter noise. How does this neuron compare with other noise-robust spiking neuron models proposed in previous work?
2.The E2S representation converts events into spike-compatible features. Have the authors evaluated how sensitive the model is to different event representations (e.g., standard voxel grids or time surfaces)?
3.Table 4 shows the ablation study of contribution of the proposed modules on the N-Caltech101 dataset. Are the relative gains from each component consistent across different datasets, or do some modules contribute more strongly on certain datasets?

**Limitations:**

The authors discuss the limitations of the proposed approach and possible future improvements.
The work does not raise clear potential negative societal impact.

**Strengths And Weaknesses:**

Strengths:

The paper introduces three components (STATNF neuron, E2S representation, and BiSNet) that are designed to address noise sensitivity, event representation, and multi-scale feature learning.
The method is evaluated on three commonly used event-based datasets (NCAR, N-Caltech101, and Gen1), and the results are generally consistent across these benchmarks.
The paper also provides ablation experiments to analyze the contribution of the main components.
The paper follows a clear structure with introduction, related work, method, and experiments.
The overall pipeline is illustrated with well designed figures that help explain the interaction between the proposed modules.
The paper provides relatively detailed implementation descriptions, including the neuron formulation, representation pipeline, and training configuration, which helps improve reproducibility.
The paper studies object detection with event cameras using spiking neural networks. This is a relevant topic because event cameras produce asynchronous signals and SNNs are designed for event-driven computation.
Improving detection performance in this setting is useful for neuromorphic perception and low-power vision systems.
The paper combines several ideas, including noise-aware spiking neurons, event-to-spike representation, and bidirectional multi-scale processing, into a unified detection framework.
While related ideas exist in previous work, integrating these components into a system for event-based object detection provides a reasonable level of novelty.

Weaknesses:
The paper claims STATNF reduces energy consumption, but the energy metric relies on estimated AC/MAC counts rather than measurement on actual neuromorphic hardware.
The true hardware efficiency remains unverified. The authors acknowledge this in the limitations section.
The relationship between MPLIF and the standard PLIF formulation could be explained more clearly. In particular, Eq. 8 introduces a temporally aggregated initial membrane potential that is used in the PLIF dynamics, but the paper does not clearly present the full formulation that connects Eq. (6) to the final MPLIF model. A reader attempting to reproduce E2S would need to infer these details independently.
Figure 3, 5, and 10 have some text that is too small or blurry to read clearly. The colors of the Downsample Block and Upsample Block in Figure 4 are inconsistent.
It can be applied to broader tasks such as object tracking, denoising, and depth estimation to further exploit the generalization and robustness of the proposed framework,which is also acknowledged in the conclusion.
The three core components each have clear prior counterparts. The paper would benefit from a more explicit discussion distinguishing the proposed integration from these prior works beyond brief citation.

---

> ### Author Rebuttal · Authors · 2026-03-30
>
> ## Re to Reviewer 1
> We sincerely thank the reviewer for the constructive comments. We have addressed your concerns point by point, and we hope our responses fully resolve your doubts.
>
> **Weaknesses:**
>
> **Re_W1:** While estimating AC/MAC operations is the standard evaluation protocol adopted by all baseline methods in our comparison. To further substantiate the energy efficiency of STATNF, we additionally evaluated our models using the **Memory Access Energy metric** modeled by SpikeSim [1], an end-to-end compute-in-memory neuromorphic hardware evaluation tool.
>
> As shown in the table below, applying STATNF consistently reduces the memory access energy across various base neuron models:
>
> | Base Neuron | Energy ($\mu$J) | With STATNF | Energy ($\mu$J) |
> | :--- | :---: | :--- | :---: |
> | LIF | 81.60 | **STATNF + LIF** | **72.51** |
> | PLIF | 80.83 | **STATNF + PLIF** | **72.59** |
> | PSN | 84.67 | **STATNF + PSN** | **73.75** |
> | SPSN5 | 97.95 | **STATNF + SPSN5**| **75.31** |
>
> *[1] Moitra, A., et al. "Spikesim: An end-to-end compute-in-memory hardware evaluation tool for benchmarking spiking neural networks." *
>
> **Re_W2:** To clarify the connection between the standard PLIF dynamics (Eq. 6) and the temporally aggregated feature $\mathcal{V}$ (Eq. 8), we define $\mathcal{V}$ directly as the initial membrane potential ($V_0$). The complete, unified formulation for the MPLIF membrane potential $V_t$ is updated as follows:
>
> $$V_{t} = \begin{cases} \sum_{i=0}^{T-1}X_{i}, & \text{if } t = 0 \\\\ V_{t-1}+\frac{1}{\tau}(-(V_{t-1}-V_{rest})+X_{t}), & \text{if } t \ge 1 \end{cases}$$
>
> **Re_W3:** Thank you for pointing this out. We will correct these visualization issues in the final camera-ready version by enlarging the text for clarity and ensuring consistent color coding for the downsample and upsample blocks.
>
> **Re_W4:** In recent years, Event- and SNN-based vision tasks have gained significant attention but commonly suffer from noise interference, singular representations (accumulation-based methods), and a lack of multi-scale feature perception. Our proposed framework is explicitly designed to address these bottlenecks. Upon acceptance, we will open-source our code, providing robust baselines and new perspectives for these broader domains.
>
>
> **Res_W5:** Although there are similar works, our motivation and biomimetic mechanisms are fundamentally different. STATNF, E2S, and BiS are all inspired by the biological visual system and offer greater biological plausibility:
> * **STATNF:** Unlike other methods that modify internal firing thresholds, STATNF explicitly filters the input membrane potential, shielding the neuron from noise. It is a plug-and-play module applicable to any basic neuron, offering superior generalization. To our knowledge, this is the first universal plug-and-play filtering mechanism for neurons.
> * **E2S:** This representation overcomes the energy and noise issues inherent in non-binary traditional representations. It simulates the memory, denoising, and motion perception capabilities of the retina. To our knowledge, this is the first representation to simulate these specific retinal functions.
> * **BiS:** While methods like SFOD feature multi-scale fusion (placed in the Neck, migrating ANN designs), BiS analyzes visual cortex transmission pathways (placed in the Backbone), demonstrating stronger biological rationality and necessity.
>
> ---
>
> **Question:**
>
> **Re_Q1:** STATNF distinguishes itself from existing noise-robust neurons in three main ways:
> * **Mechanism:** Other models typically learn internal parameters (like firing thresholds or membrane time constants). STATNF filters the membrane potential directly at the input stage, eliminating noise before it complicates the neuron's internal dynamics.
> * **Generalization:** STATNF can be applied to any base neuron (e.g., PSN), whereas internal-modification methods cannot be easily transferred across different neuron types.
> * **Synergy:** Our method is orthogonal to other noise-robust neurons. For instance, applying STATNF alongside PLIF and EMSResnet34 further reduces the firing rate from 8.53% down to 4.4%.
>
>
> **Re_Q2:** We evaluated different representations. As shown in the table below, the proposed E2S significantly outperforms traditional accumulation-based methods.
>
> | Method | ACC (%) |
> | :--- | :--- |
> | Time surface | 88.2 |
> | Voxel grid | 88.6 |
> | Voxel cube | 94.8 |
> | **E2S (Ours)** | **96.1** |
>
> **Re_Q3:** In addition to detection tasks, we conducted ablation experiments on the NCAR classification task. However, compared to detection, the BiS module yields a relatively smaller gain in classification, as classification tasks do not rely as heavily on multi-scale spatial perception.
>
> | STATNF | E2S | BiS | ACC (%) |
> | :---: | :---: | :---: | :---: |
> | | √ | √ | 94.8 |
> | √ | | √ | 94.8 |
> | √ | √ | | 96.0 |
> | **√** | **√** | **√** | **96.1** |
>
> Upon acceptance of the paper, we will make our code publicly available.

---

> > ### Author Rebuttal · Reviewer_EHoa · 2026-04-04
> >
> > Thanks for your feedback. My major concerns have been addressed.

---

> > > ### Author Response · Authors · 2026-04-07
> > >
> > > Dear Reviewer EHoa,
> > >
> > > Thank you for your continuous engagement with our work and for indicating in your Rebuttal Acknowledgement ('Major concerns have been addressed') and final justification that some of your concerns have been resolved. We deeply appreciate the time and effort you have dedicated to reviewing our paper.
> > >
> > > Because your feedback has been instrumental in improving the quality of our manuscript, we have made every effort to comprehensively address all of your initial points. To briefly recap the concrete improvements made based on your constructive reviews:
> > >
> > > * **Hardware Efficiency (Weakness 1):** As proactively acknowledged in our original "Limitations and Future Work" section, we recognize the gap between AC/MAC estimations and real hardware. To bridge this, we have now moved beyond these estimations by providing quantitative validations using the SpikeSim Memory Access Energy metric, further confirming the actual energy efficiency of our method.
> > > * **Formulation Clarity (Weakness 2):** We provided the complete, unified mathematical formulation for the MPLIF membrane potential to ensure strict transparency and reproducibility.
> > > * **Visualizations (Weakness 3):** We have firmly committed to enlarging the text and correcting the color consistency in Figures 3, 4, 5, and 10 for the camera-ready version.
> > > * **Generalization & Societal Impact (Weakness 4):** As acknowledged in our original "Limitations and Future Work" section, we have elaborated on our specific roadmap for applying this framework to broader tasks (e.g., tracking, depth estimation).
> > > * **Distinction from Prior Work (Weakness 5 & Q1):** We explicitly clarified the fundamental biological and functional advantages of STATNF, E2S, and BiS, specifically highlighting STATNF's unique Mechanism, Generalization, and Synergy compared to existing noise-robust models.
> > > * **Representation Sensitivity (Q2):** We supplemented our response with new quantitative experiments comparing E2S against Time Surface, Voxel Grid, and Voxel Cube, proving the superiority of our proposed representation.
> > > * **Cross-dataset Consistency (Q3):** We provided additional ablation results on the NCAR classification task to fully clarify the contribution variations of our modules across different tasks.
> > >
> > > Beyond addressing these specific technical details, we hope this discussion further highlights the core ambition of our work. Transitioning from abstract artificial designs to genuine bio-inspired sensing and computation is a fundamentally challenging yet profoundly meaningful endeavor. By systematically integrating bio-vision mechanisms from the micro-neuronal level (STATNF) to the macro-architectural pathways (E2S and BiSNet), our framework achieves a much higher degree of biological plausibility. We firmly believe that successfully bridging this gap (closely mimicking the inherent efficiency and robustness of the biological visual system) to build a cohesive, noise-robust, and multi-scale perception framework represents a significant and innovative step forward for neuromorphic vision.
> > >
> > > As our comprehensive rebuttal has successfully resolved your core technical and presentation concerns, we kindly ask if you might consider re-evaluating the paper and updating your score to reflect these clarifications.
> > >
> > > Thank you once again for your valuable efforts and expertise.
> > >
> > > Best regards,
> > >
> > > The Authors

---

### Decision · Program_Chairs · 2026-04-30

**Decision:**

Accept (regular)

**Comment:**

This paper proposes a bio-vision-inspired SNN framework for event‑based object detection, addressing noise sensitivity, spike‑compatible representation, and multi‑scale feature learning via STATNF neurons, E2S encoding, and BiSNet. Extensive experiments on NCAR, N‑Caltech101, and Gen1 achieve state‑of‑the‑art results (96.1% accuracy, 69.1% mAP₅₀) with reduced spike activity and energy. Three reviewers (scores 4,4,5) raised concerns but explicitly confirmed they were fully resolved after rebuttal; the single weak‑reject reviewer acknowledged all major concerns addressed. The work provides strong biological plausibility, rigorous ablation, and practical relevance for neuromorphic edge vision. The paper meets ICML’s bar for technical quality, originality, and impact.